

# Sensitivity of ice production estimates in Laptev Sea polynyas to the parameterization of subgrid-scale sea-ice inhomogeneities in COSMO-CLM

Oliver Gutjahr[1], Günther Heinemann[1], Andreas Preußer[1], Sascha Willmes[1], and Clemens Drüe[1]

[1]Department of Environmental Meteorology, University of Trier, Behringstraße 21, 54296 Trier, Germany

*Correspondence to:* Oliver Gutjahr (gutjahr@uni-trier.de)

**Abstract.** A tile-approach (TA) for the calculation of the energy balance over fractional sea ice was implemented into the standard version of the COSMO-CLM (CCLM) model. The tile-approach accounts for subgrid-scale energy exchange within polynyas and leads, which are neglected in the CCLM standard version. We perform six simulations for the area of the Laptev Sea at a horizontal resolution of $5\,\mathrm{km}$ for the winter season 2007/08 (Nov.-Apr.) with different grid-scale and subgrid-scale ice thicknesses within polynyas. A reference run without TA assumes a grid-scale ice-thickness of $10\,\mathrm{cm}$ within polynyas (derived as the mean thin-ice thickness in the Laptev polynyas from satellite data). Three sensitivity runs were performed for $10\,\mathrm{cm}$ grid-scale ice thickness and subgrid-scale open water, thin-ice of $1\,\mathrm{cm}$, and $10\,\mathrm{cm}$ thickness. Two runs use a grid-scale ice thickness of $50\,\mathrm{cm}$ and a subgrid-scale ice thickness of $5\,\mathrm{cm}$ and $1\,\mathrm{cm}$, respectively. We analyse the sensitivity of the ice production (IP) in this winter and compare them with estimations from remote sensing methods. In addition, the impact of the surface heat exchange on the atmospheric boundary layer (ABL) is shown for a case study.

The use of the TA causes an increased heat loss over polynyas, which is up to $+109.7\,\%$ higher in the sensitivity runs compared to the reference run. This enhanced heat loss is caused by an increase of the surface temperatures and the near-surface wind speed within and above polynyas. The surface temperatures are $+6\,^{\circ}\mathrm{C}$ to $+16\,^{\circ}\mathrm{C}$ higher than in the reference simulation. The reference ice production of $29.05\,\mathrm{km}^3$ increases due to the enhanced heat loss in the sensitivity simulations by $+0.3\,\%$ to $+124.5\,\%$. The comparison of the IP with estimates from remote sensing methods remains difficult due to different assumptions on the combination with atmospheric data, turbulent transfer coefficients for heat and polynya definitions.

In summary, the consideration of subgrid-scale energy fluxes in form of the tile approach yields a more realistic representation of fractional sea ice cover. However, the impact on IP and the ABL depends strongly on the choice of the subgrid-scale ice thickness, which should be consistent with satellite-derived ice thickness distribution in polynyas.

## 1 Introduction

The rate of sea-ice growth strongly depends on the energy fluxes at the ice or ocean surface. If the total atmospheric heat flux is negative, the ocean is losing heat either directly to the atmosphere or via conduction through an existing sea-ice cover. In the former case frazil ice forms, which aggregates subsequently to a new thin-ice layer under calm conditions. In the latter case basal freezing occurs to balance this heat loss.



In the standard version (v5.0_clm1) of the regional climate model 'COnsortium for Small-scale MOdel - Climate Limited area Mode' (COSMO-CLM or CCLM; Rockel et al. (2008)), which is the climate version of the numerical weather prediction model COSMO of the German Meteorological Service (Steppeler et al., 2003), a model grid box is either assumed to be completely covered with sea ice or to be completely ice-free. However, most of the energy exchange between the ocean and the atmosphere occurs over open water or thin-ice areas, such as leads or polynyas, within an otherwise compact sea-ice cover (Smith et al., 1990; Morales Maqueda et al., 2004; Ebner et al., 2011). Although the fraction of such areas in polar oceans is relatively small during winter, they are of major importance for the heat budget and ocean circulation (Heinemann and Rose, 1990; Haid et al., 2015).

We hypothesise that neglecting subgrid-scale open water/thin-ice areas could result in an underestimation of the energy transfer and hence in an underestimation of newly grown sea ice. This underestimation affects the sea-ice budget and associated processes connected to the ocean, such as salt release and deep water formation.

The horizontal resolution of regional climate models is generally too coarse to represent leads and small polynyas explicitly. Therefore, they have to be treated as inhomogeneities of momentum and energy fluxes on a subgrid scale. Heinemann and Kerschgens (2005) investigated three approaches to account for such subgrid-scale inhomogeneities within a model grid box: the (i) aggregation, (ii) mosaic and (iii) tile-approach (TA). In the aggregation approach the parameters for the fluxes (such as roughness length or albedo) are weight-averaged over different surface types within a grid box and then the fluxes are calculated from these grid-scale means. In contrary, in the mosaic approach the fluxes are explicitly calculated on a sub-scale grid and averaged afterwards.

The TA is a simplification of the mosaic approach, considering only the percentage of different surface types but not their exact location. According to Heinemann and Kerschgens (2005) the TA provides similarly good results as the mosaic approach, but with distinctly less computation time. Thus, we decided to implement this variant. First steps in the direction of a tile-approach in CCLM were made by Van Pham et al. (2014). However, their adjustments were limited to area-weighted albedo values and to surface roughness values within a grid box that is covered with fractional sea ice. In other regional climate models, such as the Weather Research & Forecasting (WRF) Model, fractional sea ice is already a default option (Bromwich et al., 2009). However, assumptions have to be made for the subgrid-scale thin-ice thickness, since particularly in winter leads and polynyas are rarely ice-free (Willmes et al., 2011).

To improve the energy exchange over fractional sea ice in CCLM, we modified the standard version with regard to the following points: (i) we implemented the thermodynamic 2-layer sea-ice module of Schröder et al. (2011), (ii) we used daily sea-ice thickness (SIT) fields from the Pan-Arctic Ice-Ocean Modeling and Assimilation System (PIOMAS) data set (Zhang and Rothrock, 2003) as initial data, (iii) we implemented a new albedo-scheme for sea ice based on Køltzow (2007), and (iv) we implemented a tile-approach for the energy balance over fractional sea ice.

In the following we investigate the sensitivity of ice production (IP) rates in the Laptev Sea polynyas (Sibiria) on the assumptions of thin-ice thickness associated with the tile-approach. Although points (ii)-(iii) represent new modifications to CCLM as well, we accept them as the default option for our reference simulation. The sea-ice module of Schröder et al. (2011) was already successfully applied in the Laptev Sea by Ebner et al. (2011), who could show that polynyas significantly affect





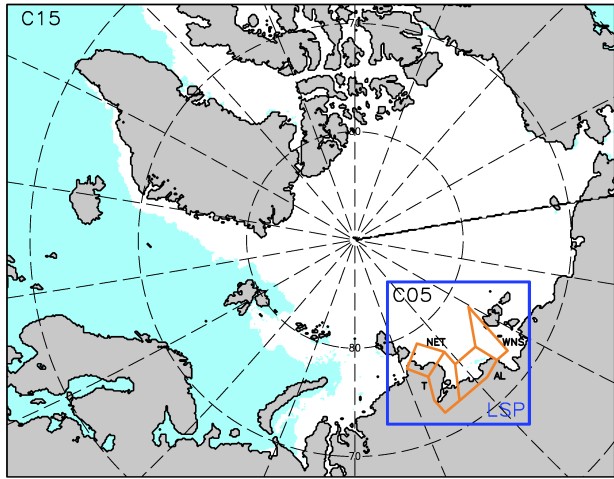

**Figure 1.** Model domains of COSMO-CLM at a horizontal resolution of 15 km (C015, whole Arctic). The study domain of the Laptev Sea polynyas (LSP) with a resolution of 5 km (C05, blue box) is shown in detail in Fig. 2. The sea-ice extent (white shaded) is from 4 January 2008.

the atmospheric boundary layer. More recently, Bauer et al. (2013) calculated sea-ice production rates for this region with an assumed thin-ice thickness of 10 cm (B10) or open water (B00) within polynyas. Their model results showed that the presence of grid-scale thin-ice affects the IP considerably.

The implementation of a TA for subgrid-scale energy fluxes constitutes, from a physical point of view, an improvement of

5    representing polynyas in regional climate models. However, it is unclear how sensitive the energy fluxes and the resulting IP are to the choice of grid-scale and subgrid-scale ice thickness. By varying the ice thickness in a sensitivity experiment, we aim to quantify these uncertainties. As a benchmark for our study we use the IP estimations of Willmes et al. (2011). We further comprise model results of Bauer et al. (2013) and derived IP from Moderate Resolution Imaging Spectroradiometer (MODIS) data.

10   This paper is structured as follows: in section 2, a short overview of the model configuration and the study region is given; in section 3 the basics of the sea-ice module are described (see details in appendix A and appendix B). Section 4 shows the calculation of sea-ice production. The model is validated with in situ data in section 5 and the effects on the atmospheric boundary layer and on ice production rates are presented in section 6 and 7 and discussed with respect to remote sensing estimates in section 8. Finally, we draw conclusions in section 9.



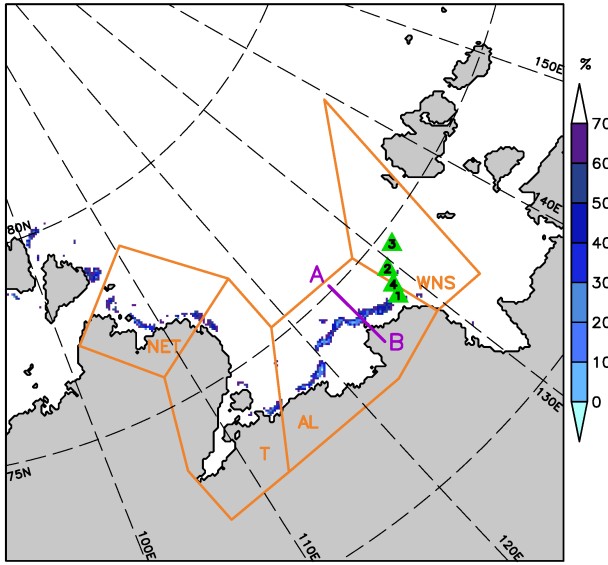

**Figure 2.** Model domain at $5\,\mathrm{km}$ resolution (C05) over the Laptev Sea (approximately $1500\,\mathrm{km} \times 1500\,\mathrm{km}$) with the sea-ice concentration from AMSR-E showing open polynyas ($\leq 70\,\%$) on 4 January 2008. Four polynya regions are superimposed as orange polygons: north-eastern Taimyr polynya (NET), the Taimyr polynya (T), the Anabar-Lena polynya (AL) and the western New Siberian polynya (WNS). A→B denotes the $214\,\mathrm{km}$ long cross-section (magenta) used in section 6. The locations of the four AWS stations are marked with green triangles.

## 2 CCLM configuration and model domains

### 2.1 Study area

The Laptev Sea polynyas (LSP), located at the Siberian coast (Fig. 1 and Fig. 2), are latent-heat polynyas or flaw polynyas (Dmitrenko et al., 2001). These polynyas are narrow, long bands of open water and/or thin-ice, which separates landfast ice
5  from seaward drifting ice on the Siberian continental shelves during winter (Dethleff et al., 1998). Under such conditions, sea water at the freezing point is directly exposed to a cold winter atmosphere resulting in intense ice formation (Dethleff et al., 1998). The Laptev Sea is a region in which a considerable amount of the total Arctic sea ice is produced (Dmitrenko et al., 2009; Dethleff et al., 1998; Willmes et al., 2011; Tamura and Ohshima, 2011; Iwamoto et al., 2014). The sea ice is subsequently transported by the Transpolar Drift System and mainly exported through Fram Strait (Krumpen et al., 2013).
10    The Laptev Sea is usually covered with pack ice from October to June and polynyas recur at quasi-stational locations (Willmes et al., 2011). Thus we subdivided this region into four polynya regions which have been already used in previous studies, e.g. by Willmes et al. (2011): the north-eastern Taimyr polynya (NET), the Taimyr polynya (T), the Anabar-Lena polynya (AL) and the western New Siberian polynya (WNS) (Fig. 2). The total area of the masks is $26.19 \times 10^4\,\mathrm{km}^2$.



**Table 1.** Overview of the performed simulations with COSMO-CLM for the winter period 2007/11–2008/04. The grid-scale thin-ice thickness (TIT) within polynyas (ice concentration: $0 < \text{SIC} \leq 70\,\%$) is shown in cm and the assumed subgrid-scale TIT is shown in parenthesis. The latter is only required if the tile-approach (TA) is used.

| Model run | $\Delta x$ | Region | TIT [cm] | Description |
|-----------|-----------|--------|----------|-------------|
| C15nt | 15 km | Arctic | 10 (-) | no TA |
| C05nt | 5 km | Laptev Sea | 10 (-) | no TA |
| C05wt0 | 5 km | Laptev Sea | 10 (0) | with TA |
| C05wt1 | 5 km | Laptev Sea | 10 (1) | with TA |
| C05wt10 | 5 km | Laptev Sea | 10 (10) | with TA |
| C05wt-50/5 | 5 km | Laptev Sea | 50 (5) | with TA |
| C05wt-50/1 | 5 km | Laptev Sea | 50 (1) | with TA |

## 2.2 Configuration of CCLM

The domain of CCLM (Fig. 1) covers the whole Arctic at a horizontal resolution of $15\,\text{km}$ (C15). CCLM was run on $450 \times 350$ grid boxes and with 42 vertical layers, whereof 16 are below $2\,\text{km}$ height. Nested within, we performed simulations for the Laptev Sea (Fig. 2) at $5\,\text{km}$ resolution (C05) with $260 \times 260$ grid boxes and 60 vertical levels, whereof 24 levels are below $2\,\text{km}$

height.

The C15 model is forced by ERA-Interim data (Dee et al., 2011) with updates to the lateral boundaries every $6\,\text{h}$. The C05 model is then forced by the output of C15 with an update frequency of $1\,\text{h}$. The models were run in a forecasting procedure for the winter period November 2007 to April 2008 ($182\,\text{days}$ in total). They were restarted every day at $18\,\text{UTC}$ and simulated the following $30\,\text{hours}$. Thereby the initial sea-ice conditions (see section 3.2) were prescribed to the sea-ice concentration and

thickness of the following day. The first 6 hours were cut off as spin-up. The simulation output ($00\text{-}23\,\text{UTC}$) was stored at a temporal resolution of 1 hour.

Surface fluxes are calculated by a bulk transfer scheme with a stability dependency (Louis, 1979) (see appendix B3). The vertical diffusion is parameterized by a level-2.5 closure scheme (Mellor and Yamada, 1974) based on a prognostic equation for turbulent kinetic energy (TKE). Radiation processes are calculated hourly using the Ritter and Geleyn (1992) scheme extended

for ice-clouds. We applied a Runge-Kutta scheme of 3rd order (Wicker and Skamarock, 2002). Additionally, a fast-wave solver for sound and gravity waves was used (Baldauf, 2013). All simulations were run without spectral nudging. We assumed a grid-scale ice thickness of $10\,\text{cm}$ within polynyas, except for two sensitivity runs where $50\,\text{cm}$ have been assumed (Tab. 1, and see section 3.2).

The 15 km simulation was performed without a TA (C15nt) in order to introduce effects from the TA only through the

5 km simulations. In case of C05, we performed a reference simulation in the Laptev Sea area without a TA (C05nt) and five sensitivity simulations with the TA. For C05nt and three of the five sensitivity simulations we assumed a grid-scale ice thickness of $10\,\text{cm}$ and either assumed subgrid-scale open water (C05wt0) or a subgrid-scale TIT of 1 cm (C05wt1) or $10\,\text{cm}$



(C05wt10). The fourth and fifth sensitivity simulations were configured with a grid-scale ice thickness of 50 cm and a subgrid-scale TIT of $5\,\mathrm{cm}$ (C05-50/5) or $1\,\mathrm{cm}$ (C05-50/1). See Tab. 1 for an overview of the simulations. The assumption of 10 cm TIT originates from the fact that the mean TIT below 20 cm, derived from MODIS data, is $\approx 10\,\mathrm{cm}$ (Willmes et al., 2011). In previous studies (Schröder et al., 2011; Bauer et al., 2013; Ebner et al., 2011) this value was assumed to be the most realistic

one for the ice thickness within the polynyas. The first three sensitivity simulations investigate the effect of the TA, if even thinner ice is assumed. The C05-50/5 and C05-50/1 runs are motivated by the fact that fractional sea-ice cover in the marginal ice zone consists of thicker ice floes (detected by microwave satellite sensors), and thin-ice of $5\,\mathrm{cm}$ or $1\,\mathrm{cm}$, respectively, which is not detected by microwave sensors.

For C05nt we further assume 1 cm thin ice at polynya grid boxes where the sea-ice concentration is exactly $0\,\%$. This
assumption is motivated by the fact that open water areas particularly produce new ice and are hence rarely free of ice.

## 3   The two-layer thermodynamic sea-ice module

### 3.1   Basic module

In this section the sea-ice module (Fig. 3) is briefly described. The module considers a snow and sea ice layer and was described and originally implemented in the COSMO model by Schröder et al. (2011). It is based on the module of Mironov et al. (2012).
For this study it is reimplemented within the version 5.0_clm1 of CCLM extended with the Køltzow sea-ice albedo scheme (see appendix A). More important for this study is the implementation of a tile-approach for the surface energy balance over fractional sea ice (see appendix B). The module and hence sea-ice growth calculation is only applied to grid boxes with an initial sea-ice cover. Formation of grease ice in open water is not parameterized in CCLM, which is even a difficult task for stand-alone sea-ice ocean models. Nevertheless, a more sophisticated parameterization has been recently developed by
Smedsrud and Martin (2015). For this reason we calculate sea-ice production in a post-processing step (see section 4).

The module assumes a constant ocean/ice interface temperature of $T_{oi} = -1.7\,^{\circ}\mathrm{C}$, i.e. $T_{oi}$ is not dependent on salinity. A temperature of $-1.7\,^{\circ}\mathrm{C}$ assumes approximately a salinity of $31.1\,\mathrm{PSU}$. The module ignores turbulent heat fluxes from the ocean at the lower boundary. Heat conductivity parameters are $2.3\,\mathrm{W\,m^{-1}K^{-1}}$ for sea ice and $0.76\,\mathrm{W\,m^{-1}K^{-1}}$ for snow. The module assumes a snow cover of $h_s = 0.1\,\mathrm{m}$ if the ice thickness exceeds a threshold $h_i > h_c$ with $h_c = 0.2\,\mathrm{m}$.

### 3.2   Sea-ice concentration and thickness for initial conditions

The sea-ice concentration (SIC) is prescribed from Advanced Microwave Scanning Radiometer-Earth Observing System (AMSR-E) data (Spreen et al., 2008), provided by the University of Bremen. The original data sets are available on a daily basis at a horizontal resolution of 6.25 km. In order to use them for CCLM, we interpolated the SIC fields onto the C15 and C05 grid, respectively, by a bilinear approach for every simulation day. All grid boxes with SIC $\leq 70\,\%$ are treated as polynyas
(Massom et al., 1998; Adams et al., 2011; Preußer et al., 2015a). Realistic polynya areas are retrieved by using this threshold, as shown by Adams et al. (2011) in comparison to a polynya signature simulation method (Markus and Burns, 1995).





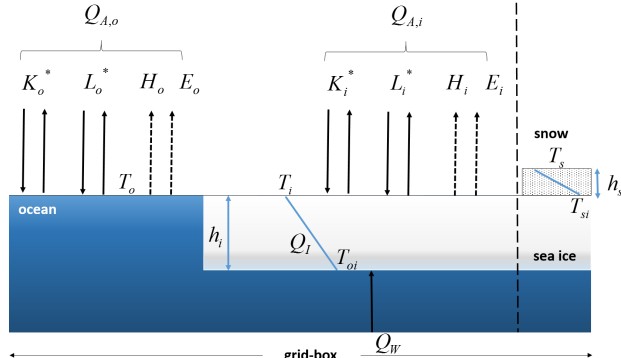

**Figure 3.** Scheme of the modified two-layer thermodynamic sea-ice module of Schröder et al. (2011), extended with a tile-approach for fractional sea ice. Sea ice is distinguished as bare ice or as snow covered ice (with $h_s = 0.1\,\mathrm{m}$ snow depth if sea-ice thickness $h_i > 0.2\,\mathrm{m}$). The subgrid-scale open ocean fraction is either ice-free (C05wt0) or assumed to be covered with $1\,\mathrm{cm}$ (C05wt1, C05-50/1), $5\,\mathrm{cm}$ (C05-50/5) or $10\,\mathrm{cm}$ thin-ice (C05wt10). If the index $k$ denotes either sea-ice ($i$) or ocean ($o$), then $Q_{A,k}$ is the total atmospheric heat flux, $K_k^*$ is the net shortwave and $L_k^*$ the net longwave radiation. $H_k$ and $E_k$ are the sensible and latent heat fluxes. $T_k$ is the surface temperature, $h_i$ the ice thickness, $T_{oi}$ the ice-ocean interface temperature and $T_{si}$ the snow-ice interface temperature. $Q_I$ denotes the conductive heat flux through the ice and $Q_W$ the turbulent heat flux from the oceanic mixed layer into the ice.

Sea-ice thickness (SIT) is taken from the Pan-Arctic Ice-Ocean Modeling and Assimilation System (PIOMAS) data set (Zhang and Rothrock, 2003). The PIOMAS data are available at a daily basis with a mean grid spacing of about 25 km (Hines et al., 2015). These daily fields were masked with the daily SIC fields to obtain consistent sea-ice extents. Thereby sea ice outside the AMSR-E mask was removed and grid boxes which were ice-free in the daily PIOMAS fields but covered with ice in the mask were assigned with an interpolated SIT from a nearest neighbour method.

Schweiger et al. (2011) state that PIOMAS seems to overestimate thin-ice thickness and underestimates thicker ice. Nevertheless, the overestimation should not be problematic in our application, since we have to set TIT for daily fields according to AMSR-E data. Underestimations of thicker ice is of a minor concern to our study due to the focus on areas with thin ice. Using this setup, the sea-ice thickness fields are much more realistic than in previous studies, where a constant thickness of $1\,\mathrm{m}$ was assumed outside polynyas (Ebner et al., 2011; Schröder et al., 2011; Bauer et al., 2013).

# 4   Estimation of sea-ice production

In accordance to previous model or satellite-based studies, the sea-ice production (IP) was calculated in a post-processing step using the energy balance (Bauer et al., 2013; Ebner et al., 2011; Willmes et al., 2011). This approach assumes that if the water within a polynya is at the freezing point, all energy loss to the atmosphere through the ocean surface is compensated by





**Table 2.** Overview of the four automatic weather stations (AWS) with hourly measurements which were deployed during the Transdrift XIII-2 expedition from 11–29 April 2008 (Heinemann et al., 2009). See the location of the AWS stations in Fig. 2.

| Station | Location | Measured period (UTC) |
| --- | --- | --- |
| AWS1 | 128.16 °E 73.80 °N | 11 Apr. 2008 07:00 – 26 Apr. 2008 12:00 |
| AWS2 | 129.32 °E 74.39 °N | 12 Apr. 2008 04:00 – 29 Apr. 2008 03:00 |
| AWS3 | 131.25 °E 74.67 °N | 14 Apr. 2008 06:00 – 29 Apr. 2008 01:00 |
| AWS4 | 128.61 °E 74.05 °N | 24 Apr. 2008 06:00 – 28 Apr. 2008 02:00 |

freezing. Hence sea-ice growth only occurs if the total atmospheric energy flux over ice (index $k = i$) or ocean (index $k = o$) $Q_{A,k} = K_k^* + L_k^* + H_k + E_k$ is negative, i.e. the ocean looses heat:

$$\frac{\partial h_i}{\partial t} = -\frac{Q_{A,k}}{\rho_i \cdot L_f}, \tag{1}$$

with $h_i$ the sea-ice thickness, $\rho_i = 910 \, \text{kg m}^{-3}$ the density of sea ice and $L_f = 0.334 \times 10^6 \, \text{J kg}^{-1}$ the latent heat of fusion. We restricted this estimation to the four polynya areas in the Laptev Sea (see Fig. 2), which are identical to those of Willmes et al. (2011). Hence, direct comparisons of our results with estimations from remote sensing are possible.

We further calculated the IP using the MOD/MYD29 sea-ice surface temperature product (Hall et al., 2004; Riggs et al., 2006) derived from MODIS Terra and Aqua data. In combination with ERA-Interim data (2 m) temperature, dew point temperature, horizontal wind components and pressure at mean sea level), an energy balance model (e.g. Yu and Lindsay, 2003; Adams et al., 2013; Preußer et al., 2015b, a) was applied to derive thin-ice thicknesses up to 0.2 m at a horizontal resolution of about 2 km. We refer to this estimation as MODIS2km. The turbulent fluxes of sensible and latent heat were calculated by an iterative bulk approach (Launiainen and Vihma, 1990) based on the Monin-Obukhov similarity theory. Thereby, the turbulent exchange coefficient $C_H$ is variable in time. Shortwave radiation is not considered as the method is restricted to wintertime. This method is only applicable to clear sky conditions, as clouds and fog impede an estimation of sea-ice surface temperature (Riggs et al., 2006). Therefore the number of useful swaths per day is variable.

Cloud-induced gaps in our daily sea-ice surface temperature and thin-ice thickness composites were filled by a spatial feature reconstruction procedure (Paul et al., 2015; Preußer et al., 2015a). This method interpolates information of previous and subsequent days to fill gaps caused by cloud-cover. Based on these corrected composites and using the method described in Preußer et al. (2015b), ice production rates were calculated for each pixel with an ice thickness $\leq 0.2 \, \text{m}$, i.e. for polynya areas.





**Table 3.** Statistical comparison of $2\,\mathrm{m}$ temperature, $10\,\mathrm{m}$ wind speed ($3\,\mathrm{m}$ in case of the AWS) and net radiation ($K^* + L^*$) of the four AWS and the C05 simulations. Hourly means are denoted by $\bar{x}$ and standard deviations are denoted by $\sigma$. The Pearson correlation coefficient ($r$) was calculated with the AWS and C05 time series. The critical correlation coefficient ($\alpha = 5\,\%$), which is depending on the sample size of the AWS time series, is between 0.1 and 0.2. In addition, the resulting p-values ($p$) of two-sided t-tests ($\alpha = 5\,\%$) are shown. Significant differences are marked with *. Data pairs with missing values in the AWS data or where the sea-ice concentration is $< 95\,\%$ were removed prior to the analyses.

| Data | 2 m temperature (°C) | | | | 10 m wind speed (m s$^{-1}$) | | | | net radiation (W m$^{-2}$) | | | |
|---|---|---|---|---|---|---|---|---|---|---|---|---|
| | $\bar{x}$ | $\sigma$ | $r$ | $p$ | $\bar{x}$ | $\sigma$ | $r$ | $p$ | $\bar{x}$ | $\sigma$ | $r$ | $p$ |
| AWS1 | -20.44 | 3.24 | - | - | 3.38 | 1.55 | - | - | -17.87 | 31.70 | - | - |
| C05nt | -20.49 | 1.95 | 0.80 | 0.83 | 3.41 | 1.57 | 0.75 | 0.83 | -30.21 | 26.01 | 0.79 | < 0.01* |
| C05wt10 | -20.60 | 1.91 | 0.79 | 0.48 | 3.42 | 1.56 | 0.75 | 0.74 | -30.18 | 26.11 | 0.80 | < 0.01* |
| C05wt1 | -20.54 | 1.92 | 0.79 | 0.66 | 3.46 | 1.58 | 0.75 | 0.55 | -30.05 | 26.25 | 0.80 | < 0.01* |
| C05wt0 | -20.52 | 1.92 | 0.79 | 0.73 | 3.48 | 1.58 | 0.75 | 0.48 | -30.36 | 26.22 | 0.81 | < 0.01* |
| C05-50/5 | -20.58 | 1.91 | 0.79 | 0.55 | 3.43 | 1.57 | 0.75 | 0.68 | -30.12 | 26.14 | 0.80 | < 0.01* |
| C05-50/1 | -20.53 | 1.92 | 0.79 | 0.68 | 3.46 | 1.59 | 0.75 | 0.56 | -29.90 | 26.35 | 0.80 | < 0.01* |
| AWS2 | -19.50 | 3.50 | - | - | 2.63 | 1.33 | - | - | -10.37 | 37.23 | - | - |
| C05nt | -20.40 | 2.23 | 0.80 | < 0.01* | 3.33 | 1.31 | 0.69 | < 0.01* | -28.61 | 24.32 | 0.73 | < 0.01* |
| C05wt10 | -20.54 | 2.15 | 0.78 | < 0.01* | 3.34 | 1.33 | 0.69 | < 0.01* | -28.36 | 24.97 | 0.75 | < 0.01* |
| C05wt1 | -20.50 | 2.12 | 0.79 | < 0.01* | 3.35 | 1.35 | 0.69 | < 0.01* | -28.72 | 25.15 | 0.75 | < 0.01* |
| C05wt0 | -20.47 | 2.13 | 0.78 | < 0.01* | 3.36 | 1.36 | 0.68 | < 0.01* | -28.70 | 25.33 | 0.75 | < 0.01* |
| C05-50/5 | -20.52 | 2.14 | 0.79 | < 0.01* | 3.35 | 1.34 | 0.69 | < 0.01* | -28.43 | 24.92 | 0.76 | < 0.01* |
| C05-50/1 | -20.49 | 2.13 | 0.78 | < 0.01* | 3.36 | 1.35 | 0.69 | < 0.01* | -28.54 | 25.13 | 0.75 | < 0.01* |
| AWS3 | -18.91 | 5.57 | - | - | 2.75 | 1.58 | - | - | -11.76 | 30.28 | - | - |
| C05nt | -19.38 | 3.07 | 0.85 | 0.20 | 3.17 | 1.48 | 0.70 | < 0.01* | -24.20 | 27.34 | 0.67 | < 0.01* |
| C05wt10 | -19.53 | 3.06 | 0.86 | 0.09 | 3.16 | 1.46 | 0.70 | < 0.01* | -23.54 | 28.65 | 0.70 | < 0.01* |
| C05wt1 | -19.47 | 3.08 | 0.86 | 0.13 | 3.17 | 1.46 | 0.69 | < 0.01* | -23.52 | 28.76 | 0.70 | < 0.01* |
| C05wt0 | -19.44 | 3.10 | 0.87 | 0.15 | 3.17 | 1.46 | 0.69 | < 0.01* | -23.39 | 28.95 | 0.69 | < 0.01* |
| C05-50/5 | -19.50 | 3.08 | 0.86 | 0.11 | 3.16 | 1.46 | 0.70 | < 0.01* | -23.39 | 28.74 | 0.70 | < 0.01* |
| C05-50/1 | -19.48 | 3.08 | 0.86 | 0.12 | 3.17 | 1.46 | 0.70 | < 0.01* | -23.46 | 28.93 | 0.70 | < 0.01* |
| AWS4 | -13.36 | 2.29 | - | - | 4.17 | 1.99 | - | - | -13.80 | 26.17 | - | - |
| C05nt | -16.17 | 3.29 | 0.67 | < 0.01* | 4.67 | 2.36 | 0.91 | 0.12 | -35.55 | 27.63 | 0.70 | < 0.01* |
| C05wt10 | -16.31 | 3.28 | 0.65 | < 0.01* | 4.64 | 2.28 | 0.92 | 0.14 | -34.31 | 28.26 | 0.73 | < 0.01* |
| C05wt1 | -16.25 | 3.26 | 0.66 | < 0.01* | 4.72 | 2.38 | 0.91 | 0.09 | -34.21 | 28.46 | 0.74 | < 0.01* |
| C05wt0 | -16.22 | 3.25 | 0.66 | < 0.01* | 4.75 | 2.40 | 0.91 | 0.07 | -34.26 | 28.51 | 0.74 | < 0.01* |
| C05-50/5 | -16.26 | 3.29 | 0.65 | < 0.01* | 4.65 | 2.29 | 0.92 | 0.13 | -34.24 | 28.43 | 0.73 | < 0.01* |
| C05-50/1 | -16.24 | 3.28 | 0.65 | < 0.01* | 4.65 | 2.29 | 0.92 | 0.13 | -34.25 | 28.51 | 0.73 | < 0.01* |

In a sensitivity analysis of this method (without the spatial feature reconstruction), Adams et al. (2013) stated an uncertainty for the ice-thickness retrieval of $\pm 1.0\,\mathrm{cm}$, $\pm 2.1\,\mathrm{cm}$ and $\pm 5.3\,\mathrm{cm}$ for thin-ice classes of $0 - 5\,\mathrm{cm}$, $5 - 10\,\mathrm{cm}$ and $10 - 20\,\mathrm{cm}$, respectively. Therefore, we constrained our analysis to ice thicknesses $\leq 0.2\,\mathrm{m}$, as this range is regarded as sufficient to get reliable results for ice production (Yu and Rothrock, 1996; Adams et al., 2013).

5    Furthermore, we compared our results to the estimations of Willmes et al. (2011). In their study they used a constant $C_H = 3 \times 10^{-3}$ to calculate $H$ and $E$ from AMSR-E data and using MODIS thin-ice distributions and National Centers for Environmental Prediction/National Center for Atmospheric Research (NCEP/NCAR) reanalysis data ($2.5° \times 2.5°$) as atmo-



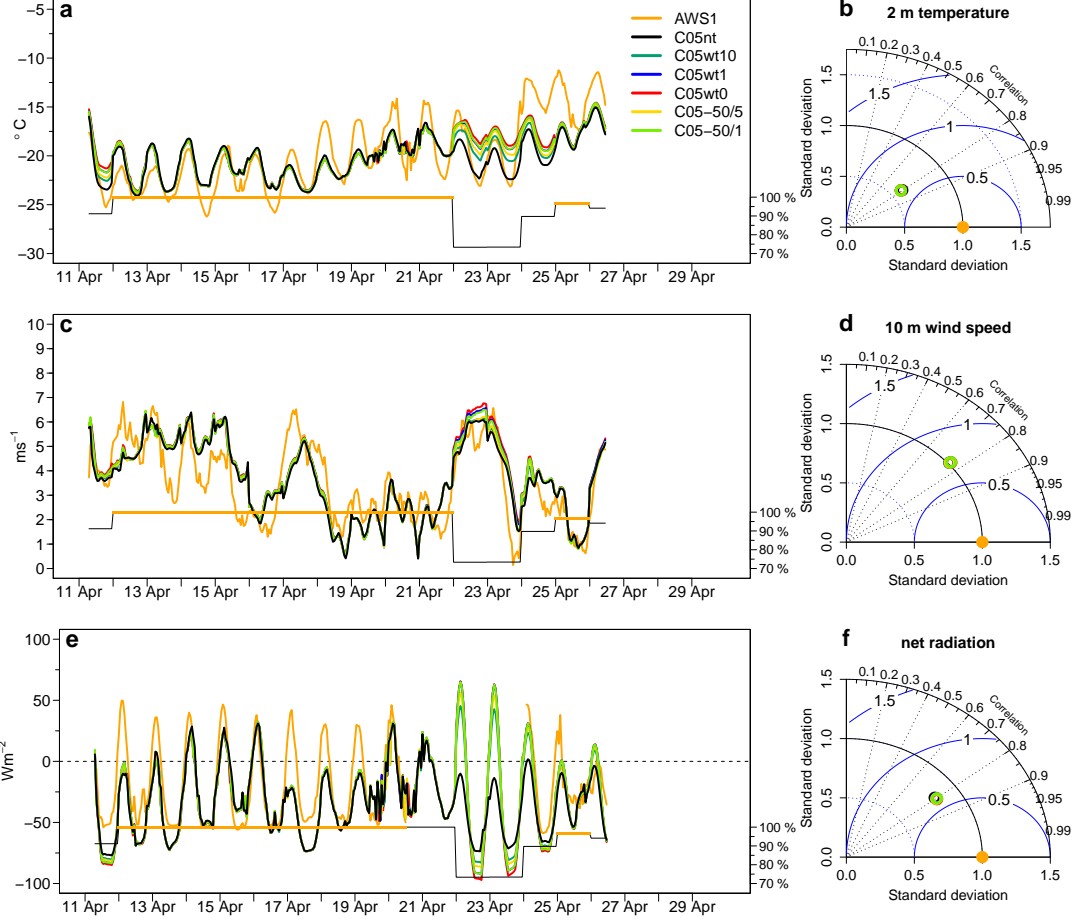

**Figure 4.** Time series comparison of AWS1 and the CCLM simulations 11-29 Apr. 2008. Time series and Taylor plots of **a,b**: 2 m temperature, **c,d**: 10 m wind speed (CCLM) and 3 m wind speed (AWS), and **e,f**: net radiation. Furthermore, the sea-ice concentration (SIC) of the CCLM grid box is shown and highlighted in orange when SIC > 95 % and no missing values occur in the AWS data. Only these values were used for the construction of the Taylor diagrams.

spheric forcing for an energy balance model. However, we omitted the most western polynya mask of their study and compare the IP only to the four remaining masks shown in Fig. 1 and Fig. 2.

We also compared our IP estimations to model-based estimations of Bauer et al. (2013). Bauer et al. (2013) conducted two COSMO simulations at 5 km horizontal resolution (without a tile-approach) for the same winter 2007/08 in the Laptev Sea. One simulation assumed a grid-scale thin-ice thickness of 10 cm within polynyas (B10) and one simulation assumed open-water (B00). Both simulations further assumed a sea-ice thickness of 1 m outside polynyas. Both simulations were forced by a 15 km COSMO simulation, which was nested within the output of the global GME model.



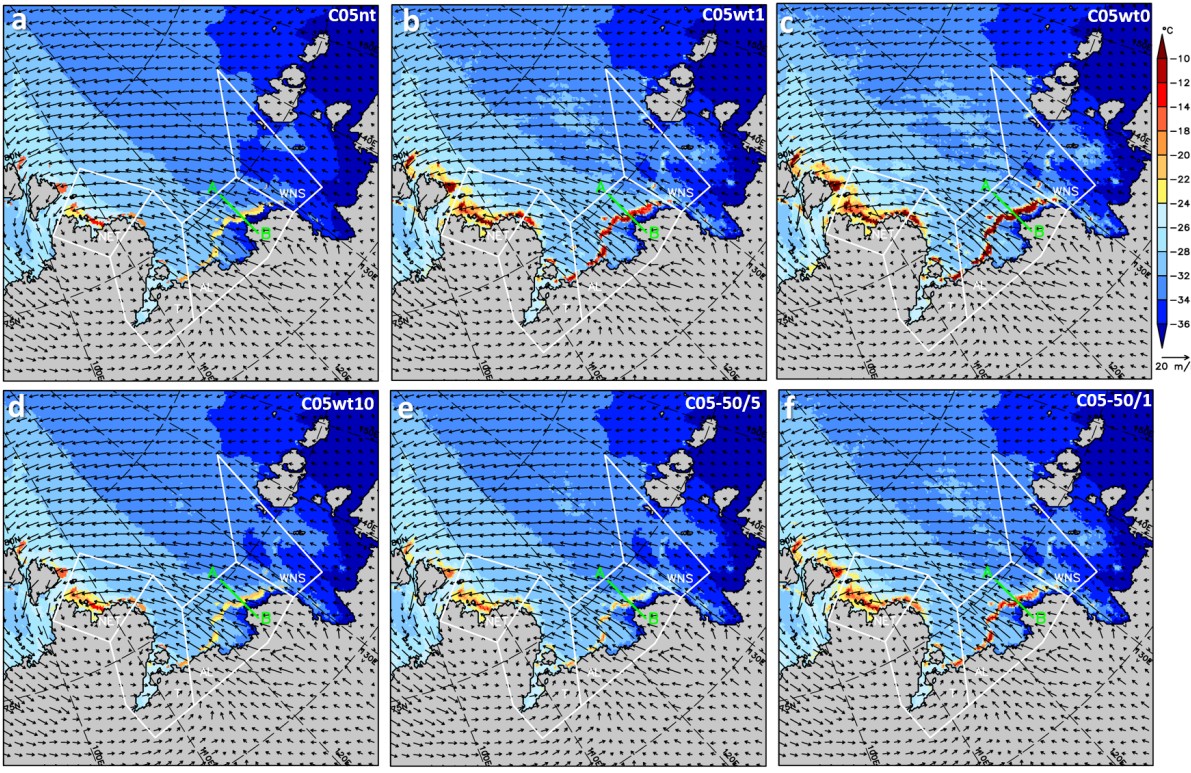

**Figure 5.** Surface temperature and $10\,\mathrm{m}$ wind field on 4 January 2008 at 15 UTC in **a**: C05nt, **b**: C05wt1, **c**: C05wt0, **d**: C05wt10, **e**: C05-50/5, and **f**: C05-50/1. The green line marks the cross-section A→B used for Fig. 7.

## 5 Verification with in situ data

The results of the five simulations were first compared to in situ data in order to verify the model configurations. During the Transdrift XIII-2 expedition from 11 April to 29 April 2008 four automatic weather stations (AWS, Tab. 2) were deployed on the fast ice of the western New Siberian Polynya (WNS, see Fig 2) (Heinemann et al., 2009). The AWS measured wind speed and direction at $3\,\mathrm{m}$ height with an accuracy of $2\,\%$ in speed and $3°$ in direction; air temperature and relative humidity at $2\,\mathrm{m}$ height with and accuracy of $0.5\,\mathrm{K}$ and $4\,\%$, and pressure with an accuracy of $1\,\mathrm{hPa}$. Furthermore, net radiation was measured by a net radiometer with an accuracy of $5\,\mathrm{W\,m^{-2}}$.

Here, we compared hourly CCLM data with the AWS data. In order to judge whether the simulations deviate significantly from the AWS data two-sided t-tests were performed ($\alpha = 95\,\%$). The statistical comparison was only performed for data pairs with no missing values and only for days when the SIC was $> 95\,\%$. This limitation is necessary because the time series of CCLM represent spatial averages of a grid box, whereas the AWS time series are point data on a solid ice cover. If the SIC of




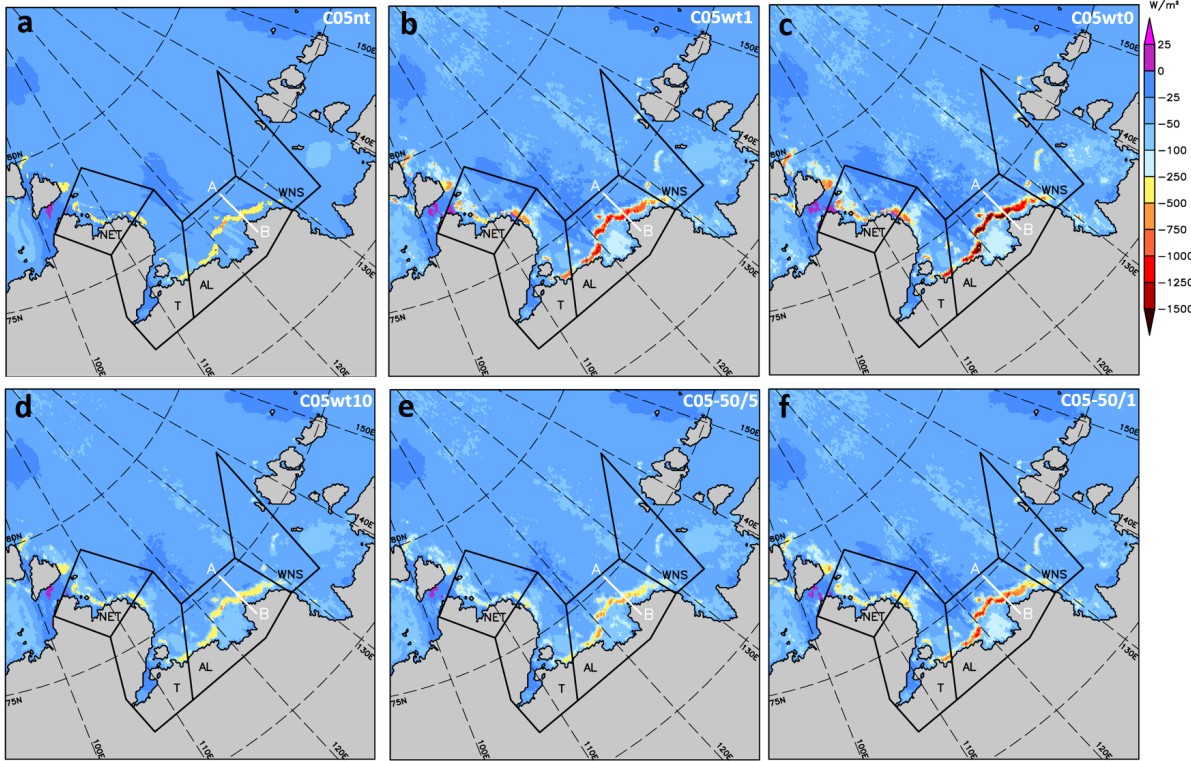

**Figure 6.** Total atmospheric energy flux on 4 January 2008 at 15 UTC in **a**: C05nt, **b**: C05wt1, **c**: C05wt0, **d**: C05wt10, **e**: C05-50/5, and **f**: C05-50/1. Negative fluxes are directed upwards. The white line marks the cross-section A→B used for Fig. 7.

CCLM is $< 100\%$ then the grid average automatically differs from the station time series, which always represent conditions at $100\%$ SIC.

The time series comparison of AWS1 and the CCLM simulations are shown in Fig. 4. Since the course of the time series of AWS2-4 and CCLM are qualitatively similar to AWS1, we only show the comparison of AWS1. A complete summary of all comparisons is compiled in Tab. 3.

In Fig. 4 the time series comparison with AWS1 is shown. Although the $2\,\mathrm{m}$ temperature of the CCLM simulations principally follows the curve progression of the AWS stations ($r \approx 0.8$, Fig. 4a), distinct differences are visible. On the first 7 days, CCLM is too warm during late evening and at night-time, while thereafter the temperature peaks at midday are underestimated. Overall, the mean temperature differs only up to $0.2\,^\circ\mathrm{C}$ (not significant), but the standard deviation is only about $60\%$ of the observation (see Tab. 3). The inter-model comparisons show no distinct differences, which is also confirmed by the Taylor plot (Fig. 4b). Larger differences only occur when the SIC becomes less than $\approx 95\%$ (21-25 Apr.). The results of the statistical





analyses show no distinct differences between the CCLM simulations: either all simulations are significant different from the observations or none.

In case of wind speed there is a good agreement ($r = 0.75$, Fig. 4c-d), although we compare $10\,\mathrm{m}$ wind speed of CCLM with measurements at $3\,\mathrm{m}$ height. The mean and standard variance are both in accordance with the AWS1 station (Tab. 3 and Fig. 4d). Reversely, this agreement implies that CCLM underestimates wind speeds at $10\,\mathrm{m}$, although we do not have reference data at $10\,\mathrm{m}$ for a verification.

Significant differences were found for the comparison of the net radiation (Tab. 3, Fig. 4e-f). Although the temporal correlation is high ($r \approx 0.8$), the mean of CCLM is about 12-13 $\mathrm{W\,m^{-2}}$ lower than the observed $-17.87\,\mathrm{W\,m^{-2}}$. These differences in the mean, and a slight underestimation of $5\,\mathrm{W\,m^{-2}}$ of the standard deviation result in significant test results. A visible inspection of Fig. 4e shows a good agreement on some days (e.g. 14-16 Apr.), but a systematic shift to more negative values on other days (e.g. 12-13 Apr.). This might be caused by errors in the cloud cover, which we unfortunately cannot compare because there are no measurements at the stations locations. The comparison with AWS2-3 shows temperature differences of about $-0.5\,^\circ\mathrm{C}$ to $-1\,^\circ\mathrm{C}$, higher wind speeds of about $+0.4\,\mathrm{m\,s^{-1}}$ to $+0.7\,\mathrm{m\,s^{-1}}$ and differences of $-13\,\mathrm{W\,m^{-2}}$ to $-14\,\mathrm{W\,m^{-2}}$ for net radiation.

Albeit some deviations CCLM is able to reproduce the basic conditions of the near-surface variables during this period. However, the reasons for these deviations need further investigation and longer time series.

## 6 Effects of the tile-approach on the atmospheric boundary layer

### 6.1 Case study on 4 January 2008

The effects of the TA are exemplified for a case study on 4 January 2008. On this day a low was located over the Taimyr peninsula in the western Laptev Sea. The large pressure gradient generated strong, prevailing off-shore winds, which caused a large opening of polynyas at the fast-ice edge in the Laptev Sea (Fig. 5a). The $10\,\mathrm{m}$ wind speed reached 10 to $15\,\mathrm{m\,s^{-1}}$ and was blowing offshore over the Anabar-Lena polynya (AL). The associated sea-ice concentrations for that day are shown in Fig. 2. Within polynyas, the SIC is about $10\,\%$ to $70\,\%$.

#### 6.1.1 Surface temperature

The surface temperatures ($T_{sfc}$) of the CCLM simulations at 15 UTC (Fig. 5) show a clear signal of the polynyas. Within the AL polynya the surface temperatures are $-22\,^\circ\mathrm{C}$ to $-24\,^\circ\mathrm{C}$ in C05nt (Fig. 5a), which is $6-16\,^\circ\mathrm{C}$ warmer than the surrounding thicker ice. Furthermore, $T_{sfc}$ is about $2\,^\circ\mathrm{C}$ warmer at the lee side than at the windward side. Stronger horizontal temperature gradients result for the NET polynyas. The first sensitivity run, C05wt10 (Fig. 5d), shows similar temperatures within polynyas and a similar wind field. Slightly warmer temperatures occur at the polynya margins, in particular in areas with $> 70\,\%$ SIC. In these areas, the subgrid-scale open water is covered with $10\,\mathrm{cm}$ thin-ice, resulting in warmer grid average temperatures compared to C05nt. If a subgrid-scale thin-ice thickness of $1\,\mathrm{cm}$ is assumed (C05wt1, Fig. 5b), the surface temperatures within





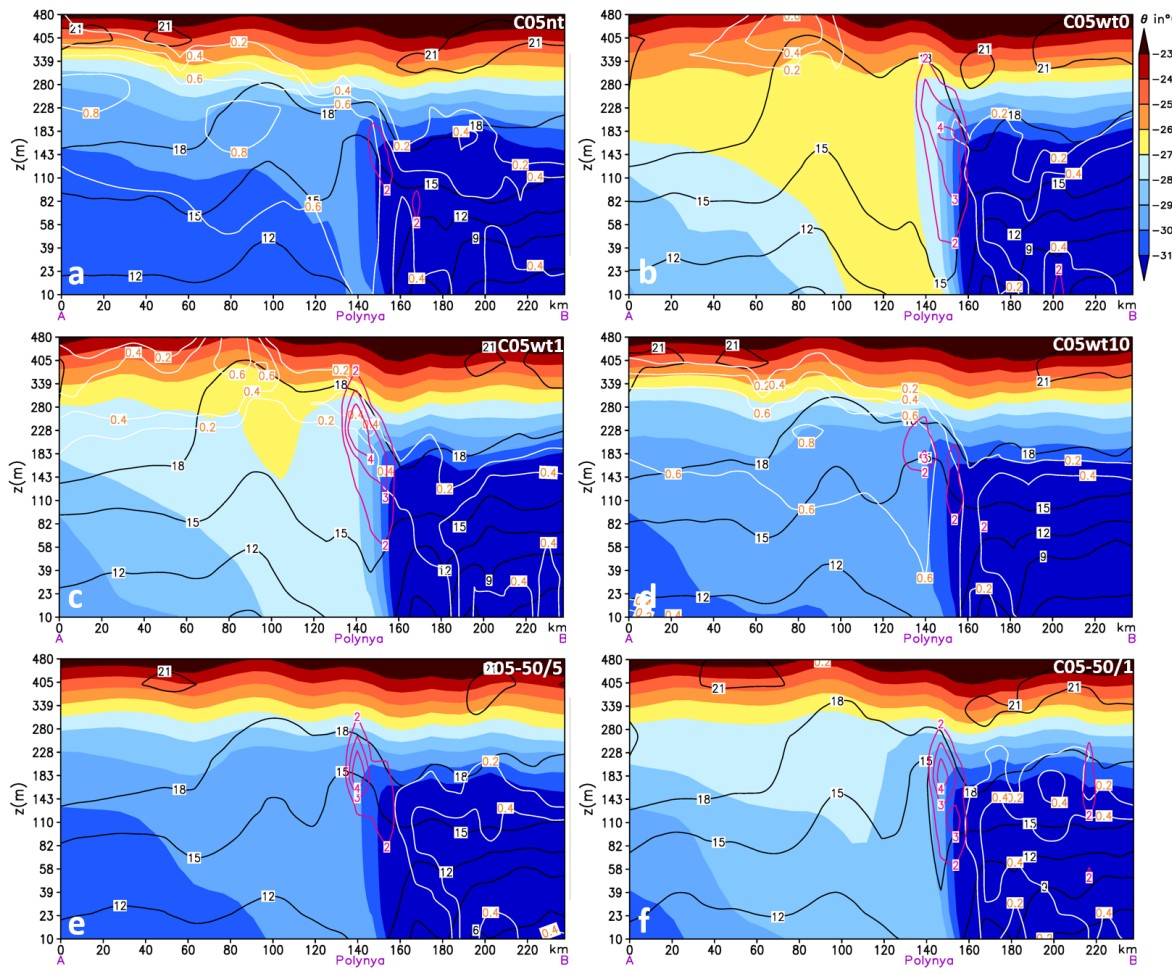

**Figure 7.** Vertical cross-sections of the potential temperature $\Theta$, horizontal wind speed (black contour lines), turbulent kinetic energy (TKE in $m^2\,s^{-2}$, magenta contour lines), and cloud fraction (white contour lines and orange labels) on 4 January 2008 at 15 UTC for **a**: C05nt, **b**: C05wt1, **c**: C05wt0, **d**: C05wt10, **e**: C05-50/5, and **f**: C05-50/1. The horizontal distance is about 240 km and the location of the cross-section A → B is shown in Fig. 2.

the polynyas become warmer than $-18\,°C$, an considerable increase of up to $+6\,°C$ compared to C05nt. A further warming occurs at the margins of the polynyas, in particular visible for the NET polynyas, and over the pack ice, where the SIC is $100\,\%$. A similar picture results for C05wt0 (Fig. 5c). The temperature within polynyas even reaches values warmer than $-10\,°C$. This increase of surface temperature is in accordance with results of Bromwich et al. (2009), who found an increase of $14\,°C$ for

5    sea-ice concentrations of about $60\,\%$ in winter. These two effects lead to an increased area where oceanic heat is exchanged



with the atmosphere. While $T_{sfc}$ of C05-50/5 (Fig. 5e) is lower than in C05nt, C05-50/1 (Fig. 5f) shows warmer $T_{sfc}$ as well, which are in between of C05wt10 and C05wt1.

### 6.1.2  10 m wind speed

The $10\,\mathrm{m}$ wind speed in C05nt (Fig. 5a) is about $14-18\,\mathrm{m\,s^{-1}}$ over the AL polynya. In C05wt0 (Fig. 5c) the wind speed increases by $2-5\,\mathrm{m\,s^{-1}}$, less for C05wt1 (Fig. 5b), C05wt10 (Fig. 5d), C05-50/5 and C05-50/1 (Fig. 5e-f). These results are in accordance with idealized studies conducted by Ebner et al. (2011) (see Fig 5c therein). Ebner et al. (2011) concluded that the increase in wind speed results in an increased net ice production, despite an increased boundary layer warming. The increase in near-surface wind speed causes a larger momentum flux (not shown) and higher energy loss from the ocean. Furthermore, although not represented in the present CCLM model, higher wind speeds increase the sea-ice drift within polynyas, so that newly formed ice is likely to drift faster, so that a strong heat loss is maintained. Both processes are expected to increase the IP. However, the latter issue has to be investigated by coupled atmosphere/sea-ice/ocean model simulations. A similar effect, although less pronounced, was simulated by C05wt1 (Fig. 5b), but no distinct deviations of the wind were found for C05wt10, C05-50/5, and C05-50/1.

### 6.1.3  Total atmospheric energy flux

The exchange of heat from the ocean to the atmosphere is summarized in the total atmospheric energy flux $Q_A$ (Fig. 6). In C05nt (Fig. 6a) $Q_A$ is slightly negative ($-25\,\mathrm{W\,m^{-2}}$ to $-100\,\mathrm{W\,m^{-2}}$) over the pack ice, and reaches about $-500\,\mathrm{W\,m^{-2}}$ over the polynyas (negative values indicate upward fluxes). Similar values within the polynyas result for C05wt10 (Fig. 7d), with slightly more negative values along the polynya margins, so that there is a transition from thin-ice to the pack ice. $Q_A$ becomes considerably more negative over polynyas in C05wt0 with $< -1000\,\mathrm{W\,m^{-2}}$ (Fig. 6c), in C05wt1 with $\approx -1000\,\mathrm{W\,m^{-2}}$ (Fig. 6b), in C05-50/5 with $\approx -750\,\mathrm{W\,m^{-2}}$ (Fig. 6e), and $-750\,\mathrm{W\,m^{-2}}$ to $-1000\,\mathrm{W\,m^{-2}}$ in C05-50/1 (Fig. 6f). Thus if the TA is used with our assumed ice thicknesses, more heat is released into the atmospheric boundary layer.

### 6.1.4  Vertical cross-sections

Fig. 7 shows cross-sections of the potential temperature $\Theta$, the horizontal and vertical wind speed, the cloud area fraction and the turbulent kinetic energy (TKE) over the AL polynya. In C05nt (Fig. 7a) $\Theta$ is about $-29\,^{\circ}\mathrm{C}$ at $10\,\mathrm{m}$ height over the polynya, about $-30\,^{\circ}\mathrm{C}$ about the pack ice, and colder than $-31\,^{\circ}\mathrm{C}$ over the fast ice. The boundary layer is stably stratified over the pack and the fast ice but well mixed and warmer over the polynya. These warm air masses reach heights of $150-200\,\mathrm{m}$ downstream the polynya. TKE values of up to $2-2.5\,\mathrm{m^2\,s^{-2}}$ are simulated over the transition from fast ice to the polynya (at km $140-160$) and downstream the polynya opening (at km $100-120$).

A similar convective boundary layer of the polynya is simulated by C05wt10 (Fig. 7d), except that the air downstream the polynya is about $1\,^{\circ}\mathrm{C}$ warmer close to the surface, because of the warmer surface temperatures at the downstream margin of the polynya and the associated enhanced heat loss. Thus, the TA increases the area and the intensity of heat loss from polynyas, so



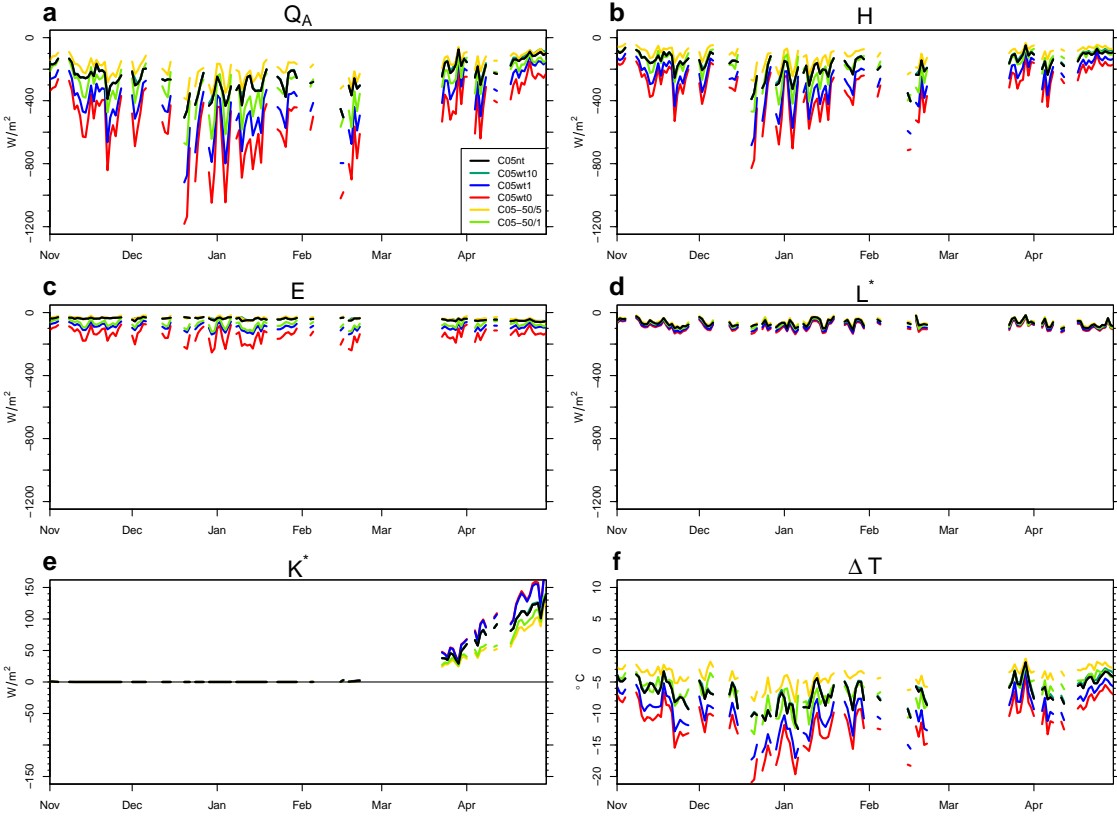

**Figure 8.** Time-series (winter 2007/08) of the daily averaged **a**: total atmospheric heat flux $Q_A$, **b**: sensible heat flux $H$, **c**: latent heat flux $E$, **d**: longwave radiation balance $L^*$, **e**: shortwave radiation balance $K^*$ and **f**: vertical temperature gradient $\Delta T$ (lowest model level (10 m) - surface), averaged over polynyas within the four polynya masks. Missing values occur if the total polynya area is $< 277\,\mathrm{km}^2$ (corresponds to 9 model grid boxes).

that the boundary layer warms compared to C05nt ($+2\,°C$ in C05wt1 (Fig. 7c) and $+3\,°C$ in C05wt0 (Fig. 7b)). The increased heat supply triggers convection above the polynya. This increases the turbulent kinetic energy (TKE) (Fig. 7) up to $> 4\,\mathrm{m^2\,s^{-2}}$ due to convection, enhances mixing and the upward vertical motion (not shown). In C05-50/5 (Fig. 7e) the convective boundary layer particularly downstream of the polynya is about $1\,°C$ warmer than in C05nt, and in C05-50/1 (Fig. 7f) about $2\,°C$. The

5  values of TKE and upward vertical motion are similar to those of C05wt1.

All simulations show a cloud fraction of $0.2 - 0.4$ in the lowest $200\,\mathrm{m}$ over the fast ice upstream of the polynya. In C05nt and C05wt10 (Fig. 7a,d) the cloud fraction over the polynya increases to $0.6$ in a layer between about $100 - 300\,\mathrm{m}$. With thinner subgrid-scale sea ice, the cloud fraction in the lower $200\,\mathrm{m}$ is less than $0.2$ (Fig. 7b-c). C05-50/5 and C05-50/01 (Fig. 7e-f) simulates no cloud formation above and downstream the polynya at all. The increased warming of the boundary layer over the




**Table 4.** Temporal means of the area-averaged energy balance, its components and the vertical gradients of temperature ($\Delta T$) and specific humidity ($\Delta q$) over polynyas (at least 9 grid boxes) for the winter period 2007/08 simulated by CCLM. The shortwave radiation $K^*$ is only of importance from March–April (begin of melting season). In parenthesis are given the percentage of the individual components of $Q_A$. Rows denoted with $\%Q_A$ show the proportion of a flux component on $Q_A$. Rows denoted with $\Delta$ show the relative change of $Q_A$ and its components of the sensitivity runs with respect to the reference run (C05nt).

| Model | $Q_A$ | $H$ | $E$ | $L^*$ | $K^*$ | $\Delta T$ | $\Delta q$ |
|---|---|---|---|---|---|---|---|
| | | | (W m$^{-2}$) | | | (°C) | ($10^{-4}$ kg/kg) |
| C05nt | $-252.5$ | $-166.7$ | $-41.0$ | $-67.4$ | $22.6$ | $-6.7$ | $-6.2$ |
| $\%Q_A$ | - | $66.0\%$ | $16.2\%$ | $26.7\%$ | $-8.9\%$ | - | - |
| C05wt10 | $-246.3$ | $-161.6$ | $-39.8$ | $-67.8$ | $22.8$ | $-6.5$ | $-5.8$ |
| $\%Q_A$ | - | $65.6\%$ | $16.1\%$ | $27.5\%$ | $-9.3\%$ | - | - |
| $\Delta$ | $-2.5\%$ | $-3.1\%$ | $-2.9\%$ | $+0.6\%$ | $+0.9\%$ | $-3.0\%$ | $-6.5\%$ |
| C05wt1 | $-414.4$ | $-267.3$ | $-91.6$ | $-82.6$ | $27.1$ | $-9.8$ | $-10.0$ |
| $\%Q_A$ | - | $64.5\%$ | $22.1\%$ | $19.9\%$ | $-6.5\%$ | - | - |
| $\Delta$ | $+64.1\%$ | $+60.3\%$ | $+123.4\%$ | $+22.6\%$ | $+19.9\%$ | $+46.3\%$ | $+61.3\%$ |
| C05wt0 | $-529.0$ | $-324.8$ | $-141.3$ | $-90.6$ | $27.7$ | $-11.6$ | $-10.4$ |
| $\%Q_A$ | - | $61.4\%$ | $26.7\%$ | $17.1\%$ | $-5.2\%$ | - | - |
| $\Delta$ | $+109.7\%$ | $+94.8\%$ | $+244.6\%$ | $+34.4\%$ | $+22.6\%$ | $+73.1\%$ | $+67.7\%$ |
| C05-50/5 | $-187.7$ | $-107.7$ | $-36.0$ | $-61.1$ | $17.1$ | $-4.1$ | $-0.3$ |
| $\%Q_A$ | - | $57.4\%$ | $19.2\%$ | $32.6\%$ | $-9.1\%$ | - | - |
| $\Delta$ | $-25.9\%$ | $-35.4\%$ | $-12.2\%$ | $-9.3\%$ | $-24.3\%$ | $-38.8\%$ | $-95.2\%$ |
| C05-50/1 | $-303.2$ | $-178.6$ | $-73.0$ | $-70.6$ | $19.0$ | $-6.5$ | $-0.7$ |
| $\%Q_A$ | - | $58.9\%$ | $24.1\%$ | $23.3\%$ | $-15.9\%$ | - | - |
| $\Delta$ | $+20.1\%$ | $+7.1\%$ | $+78.0\%$ | $+4.7\%$ | $-6.3\%$ | $-3.0\%$ | $-88.7\%$ |

polynya by the sensible heat flux divergence has a larger effect on the relative humidity than the increased moisture input by the latent heat flux. Thus the cloud formation in the model simulation is reduced for thin subgrid-scale sea ice.

## 6.2 Energy balance components for the winter period 2007/08

In order to analyse how the tile-approach affects the energy balance at the surface over the whole winter season 2007/08, we

5  compare daily means of the components of total atmospheric heat fluxes, which were spatially averaged over polynyas (Fig. 8). Thereby, at least 9 grid boxes within the polynya masks have to have a SIC$\leq 70\%$ in order to be considered in the analysis.

In principle, the processes presented in section 6.1 come into effect whenever a polynya is present. Thus, if the tile-approach is used $Q_A$ is always more negative due to the consideration of subgrid-scale energy fluxes (Fig. 8a) and more energy or heat is lost from the ocean.





Tab. 4 summarizes the temporal and spatial means and the deviations (in %) of the energy balance and its components. In C05nt the temporal mean of $Q_A$ is $-252.5\,\mathrm{W\,m^{-2}}$. Two out of the five sensitivity runs simulate less total heat loss with $-246.3\,\mathrm{W\,m^{-2}}$ or $-2.5\,\%$ (C05wt10) and $-187.7\,\mathrm{W\,m^{-2}}$ or $-25.9\,\%$ (C05-50/5). Considerably higher heat loss is simulated by the runs with subgrid-scale open-water or thin-ice between $1-5\,\mathrm{cm}$: $-529.0\,\mathrm{W\,m^{-2}}$ or $+109.7\,\%$ (C05wt0), $-414.4\,\mathrm{W\,m^{-2}}$

or $+64.1\,\%$ (C05wt1), and $-303.2\,\mathrm{W\,m^{-2}}$ or $+20.1\,\%$ (C05-50/1).

The largest contribution to $Q_A$ constitutes the sensible heat flux $H$ (Fig. 8b): $66\,\%$ in C05nt and slightly less in the sensitivity runs with $57.4\,\%$ to $65.6\,\%$. The highest impact on the sensible heat flux shows C05wt0. Here, $H$ doubles from $166.7\,\mathrm{W\,m^{-2}}$) in C05nt to $-324.8\,\mathrm{W\,m^{-2}}$. The increase is lower for C05wt1 ($+60.4\,\%$) and C05-50/1 ($+7.1\,\%$). For C05wt10 and C05-50/5 the sensible heat flux even reduces by $-3.1\,\%$ and $-35.4\,\%$.

The other components contribute much less to the heat loss. However, the partitioning changes when subgrid-scale open-water or thin-ice $< 10\,\mathrm{cm}$ is assumed. Then the contribution of $L^*$ reduces up to $-10\,\%$, except for C05wt10 and C05-50/5, but the latent heat flux ($E$) increases by up to $+10\,\%$. The reason for this is on one hand the increase of the vertical gradient of specific humidity (Tab. 4) and on the other hand the increase of the near-surface wind speed and TKE, enhancing the turbulence above the polynyas. The Bowen ratio reduces accordingly from 4.1 (C05nt and C05wt10) to 2.9 (C05wt1), 2.3 (C05wt0), 3.1

(C05-50/5), and 2.5 (C05-50/1), respectively. Shortwave radiation $K^*$ only becomes of importance in the time from March until April, when the melting season begins. Therefore, $K^*$ is small compared to the other terms.

## 7 Effects on sea-ice production

The daily sea-ice production rates were calculated for the individual polynyas as described in section 4. Here we compare the ice production of CCLM to the remote sensing estimations of Willmes et al. (2011), to estimates based on MODIS2km

(section 4), and to model results of Bauer et al. (2013).

### 7.1 Polynya area

In Fig. 9 daily polynya areas for the winter period 2007/08 are shown. According to the AMSR-E data set, which has been used to prescribe the SIC in CCLM and in the COSMO simulations of Bauer et al. (2013), large polynya events ($> 10^4\,\mathrm{km^2}$) occurred at the end of November, in January, and at the end of March. The largest opening event was observed at the end of

April. This specific event was object of research in some recent studies (Willmes et al., 2011; Bauer et al., 2013; Schröder et al., 2011; Ebner et al., 2011; Preußer et al., 2015a). The polynya areas of Willmes et al. (2011) are approximately of the same order as those used for CCLM, whereas the retrieved polynya areas from MODIS2km are considerably higher. This discrepancy is caused by different threshold definitions for polynyas and different horizontal resolutions. For the MODIS2km data, a polynya is defined as areas with thin-ice $\leq 20\,\mathrm{cm}$, as in Preußer et al. (2015b). Given that and the higher horizontal resolution it is likely

that also leads within the polynya masks, not resolved by the microwave satellite data, are contributing to the total thin-ice area and hence larger areas result. If areas of open-water outside polynyas are considered as well, then the potential area for ice production increases up to the area derived from MODIS2km data, except in the period of late November to the mid of January.





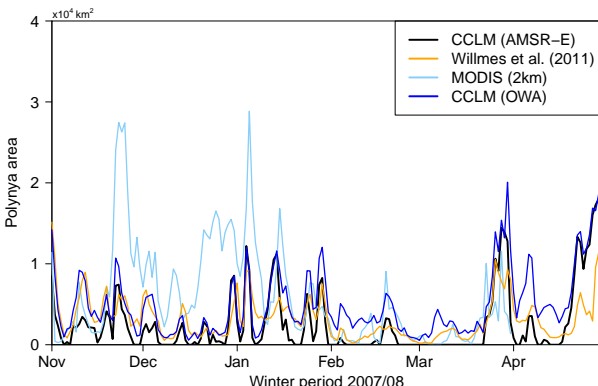

**Figure 9.** Total daily polynya area interpolated from AMSR-E onto the CCLM 5 km grid in the Laptev Sea for the winter period 2007/08 aggregated for the four polynya masks. In addition, the polynya area plus open-water area ($OWA = 1 - SIC$) is shown. Based on remote sensing the polynya areas estimated from Willmes et al. (2011) and MODIS2km data are shown. The total area of the polynya masks is $26.19 \times 10^4 \, \mathrm{km}^2$.

## 7.2 Ice production in the winter period 2007/08

The time-series of daily IP $(\mathrm{km}^3/\mathrm{day})$ are shown in Fig. 10 and the total sums for the whole winter are shown in Tab. 5. The total IP in the reference simulation (C05nt) is about $29.06 \, \mathrm{km}^3$ and only slightly higher in C05wt10 with $29.16 \, \mathrm{km}^3$ ($+0.3\%$). Both IP estimates are not significantly different to the estimate of Willmes et al. (2011). The strongest, significant increase

($p < 0.01$) in IP was simulated by C05wt0, where subgrid-scale open-water was assumed. The IP becomes $65.23 \, \mathrm{km}^3$, which constitutes a relative increase of $+124.5\%$. A significant higher IP was also simulated by C05wt1 with $49.31 \, \mathrm{km}^3$ ($+69.7\%$). A higher IP than in the reference run, but not significantly different from Willmes et al. (2011) was produced by C05-50/1 with $38.27 \, \mathrm{km}^3$ ($+38.7\%$). The only sensitivity run that produced less ice than the reference run is C05-50/5 with $25.27 \, \mathrm{km}^3$ ($-13.0\%$).

Comparing the IP of our CCLM simulations with those of Bauer et al. (2013), we found a similar ice production of C05-50/5 and B10, which are both lower than our reference run. The differences of B10 with respect to C05nt can be explained by differences in the model version, configuration, nesting chain (GME vs. ERA-Interim, different model domains). The IP of B00 is similar to C05wt1, and thus significantly higher than in Willmes et al. (2011). Of the same order is the IP derived from MODIS2km. The larger polynya areas result in a high ice production of $49.10 \, \mathrm{km}^3$ (only Nov.–Mar.), which is significantly

different ($p < 0.01$) from the estimated $33.02 \, \mathrm{km}^3$ of Willmes et al. (2011).

In general, all model simulations show a higher daily standard deviation than the estimates of Willmes et al. (2011). The temporal correlation of IP based on CCLM and the IP of Willmes et al. (2011) is $r \approx 0.6$.




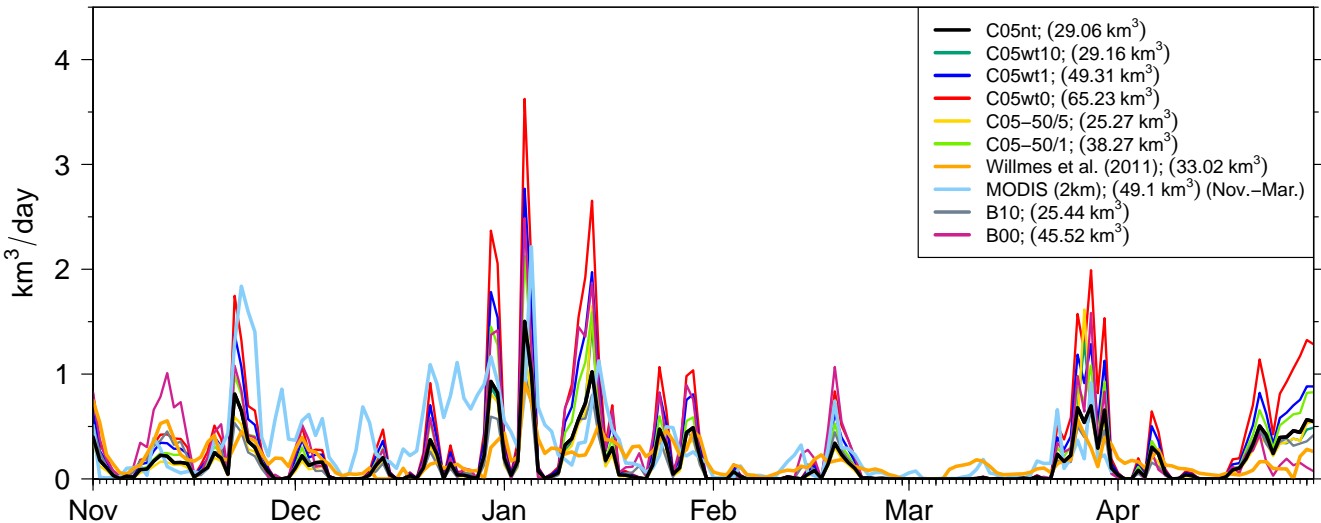

**Figure 10.** Daily sea-ice production within the Laptev Sea polynyas in the winter period 2007/08, aggregated within the four polynya masks (only considering polynyas $> 277\,km^2$ in the C05 and Bauer et al. (2013) simulations (B10, B00)). The total sea-ice production is given in parenthesis in the legend (see also Tab. 5).

Spatial maps of the total IP within polynyas in the winter 2007/08 are shown in Fig. 11. In all simulations the highest IP occurs in the NET polynyas, with rates $> 2\,m$ in C05nt (Fig. 11a) and $> 5\,m$ in C05wt0 (Fig. 11c). The lowest IP were simulated in the WNS polynyas. While the results of C05wt10 (Fig. 11d) show no distinct differences with respect to C05nt and C05-50/5 (Fig. 11e), C05-50/1 (Fig. 11f), C05wt1 (Fig. 11c), and C05wt0 (Fig. 11b) produce systematically higher IP, in particular in the AL and NET polynyas.

## 8 Discussion

Although it is not our intention to entangle all factors controlling the estimation of sea-ice production, we attempted to compile a list of influence factors, which might explain the differences we found while comparing model results with remote sensing results:

- Polynya area is affected by the definition of polynyas (SIC$\leq 70\,\%$ or $h_i < 0.2\,m$) and the horizontal resolution of the model and the satellite products.



**Table 5.** Total sea-ice production (IP) ($km^3$) in the winter period 2007/08, aggregated over polynyas within the four polynya masks (Fig. 2). The daily mean ($\bar{x}$) and standard deviation ($\sigma$) are given in $km^3$/day. The Pearson correlation coefficient ($r$) and the 95% confidence interval (CI) was calculated based on the Fisher z-transformation with the IP time series and the estimates of Willmes et al. (2011). The critical value is $r_{c,\alpha=0.05,n=182} = 0.15$ and $r$ is significant if $r > r_c$. Two-sided t-tests ($\alpha = 5\%$) were performed and the resulting p-values ($p$) values are given. Significant differences are marked with *. The results from Bauer et al. (2013) assumed an ice thickness of 10 cm (B10) or open-water (B00) within polynyas, both without a tile-approach.

| Data | total | $\bar{x}$ | $\sigma$ | $r$ [95% CI] | $p$ |
|---|---|---|---|---|---|
| C05nt | 29.06 | 0.16 | 0.24 | 0.65 [0.56;0.73] | 0.30 |
| C05wt10 | 29.16 | 0.16 | 0.26 | 0.63 [0.53;0.71] | 0.34 |
| C05wt1 | 49.31 | 0.27 | 0.42 | 0.63 [0.53;0.71] | 0.01* |
| C05wt0 | 65.23 | 0.36 | 0.57 | 0.61 [0.51;0.69] | < 0.01* |
| C05-50/5 | 25.27 | 0.14 | 0.25 | 0.56 [0.45;0.65] | 0.05* |
| C05-50/1 | 38.27 | 0.21 | 0.34 | 0.60 [0.50;0.69] | 0.30 |
| Willmes et al. | 33.02 | 0.18 | 0.16 | - | - |
| MODIS2km[1] | 49.10[1] | 0.32[1] | 0.38[1] | 0.45 [0.33;0.56][2] | < 0.01*[2] |
| B10 | 25.44 | 0.14 | 0.19 | 0.67 [0.58;0.74] | 0.02* |
| B00 | 45.52 | 0.25 | 0.40 | 0.68 [0.59;0.75] | 0.03* |

[1] Only for November - March.

[2] Comparisons with Willmes et al. (2011) were made only for November - March.

- Heat loss is affected by differences in the surface temperature, vertical temperature gradient, parameterization of the energy balance components, sea-ice thickness and properties, parameterization of the heat flux through the ice, and by the paramaterization of atmospheric turbulent fluxes. Particularly important is the horizontal resolution of the atmospheric data set and the assumptions on the turbulent exchange coefficient for heat ($C_H$). Willmes et al. (2011) assumed a constant $C_H = 3 \cdot 10^{-3}$, However, the mean values from C05 over polynyas (winter 2007/08) are about $(2.5 \pm 0.28) \cdot 10^{-3}$ (Fig. 12).Except for C05-50/5 and C05-50/1, which simulated slightly lower values of $(2.27 \pm 0.18) \cdot 10^{-3}$ and $(2.31 \pm 0.18) \cdot 10^{-3}$.

  Since the warm surface temperatures of polynyas and the resulting vertical temperature gradients are not well represented in ERA-Interim or NCEP, the usage of a high value of $C_H$ seems to partly compensate for this issue. The $C_H$ values based on MODIS data and ERA-Interim are lower than simulated by CCLM with a mean of $C_H = (2.3 \pm 0.3) \cdot 10^{-3}$. A similar PDF was derived by Adams et al. (2013), who combined MODIS and NCEP. Because of the horizontal resolution of MODIS, polynyas are represented in the surface temperature causing a larger vertical temperature gradient and hence $C_H$ values comparable to CCLM.

- Surface temperatures in remote sensing approaches also depend on the number of swaths per day, e.g. clear-sky conditions, and their distribution over the day. If not equally distributed, the surface temperature may be biased.





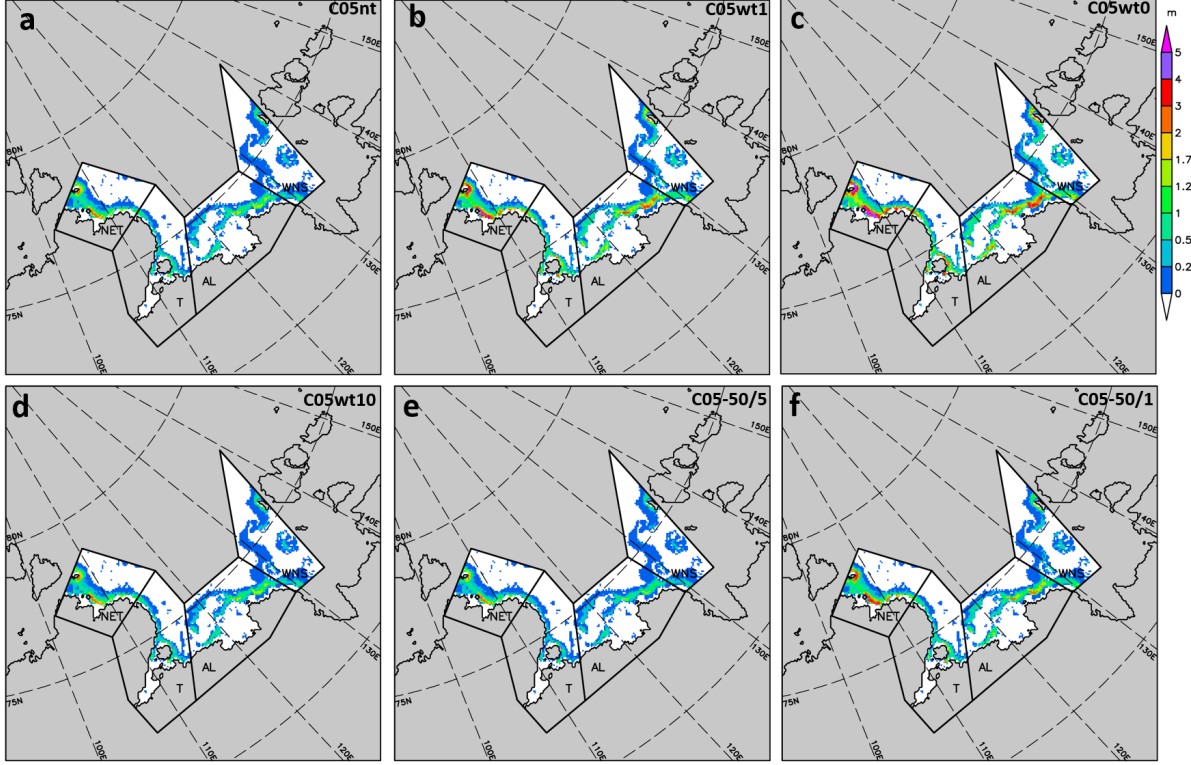

**Figure 11.** Total sea-ice production (m) within the Laptev Sea polynyas in the winter period 2007/08 simulated by **a**: C05nt, **b**: C05wt1, **c**: C05wt0, **d**: C05wt10, **e**: C05-50/5, and **f**: C05-50/1.

These influence factors together control the sea-ice production estimates and differences between model results and remote sensing. The polynya area is an obvious factor with the simple relationship: the larger the polynya area the larger the sea-ice production. In contrast, the explanations of differences in the loss of heat or energy within polynyas is manifold. In our opinion, the most relevant factors, besides polynya area, are the thin-ice thickness and the parameterizations of the turbulent heat fluxes, in particular the differences in $C_H$.

The complexity of these factors makes a comparison of model and remote sensing studies difficult. It further indicates that some assumption in the remote sensing approaches, such as a constant value for $C_H$, might be oversimplified. Furthermore, a problematic issue is the use of coarse atmospheric data sets, such as NCEP or ERA-Interim, for remote sensing approaches, if not combined with high-resolution satellite products. The horizontal resolution of such atmospheric reanalysis data sets is not sufficient to represent polynyas adequately. Thus subsequent errors, such as wrong simulations of the atmospheric boundary layer over polynyas, are the consequence. These errors are then transferred to the remote sensing approach and might result in wrong sea-ice production estimates. From a modelling point of view the question arises what reference for IP estimates should





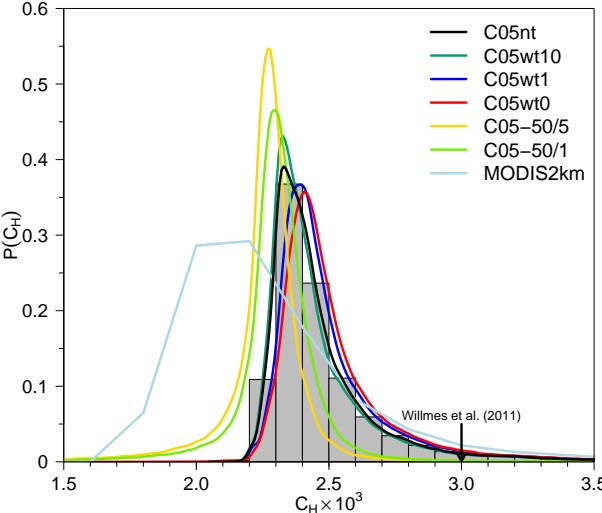

**Figure 12.** Probability density functions of the turbulent transfer coefficients for heat ($C_H$) within the Laptev Sea polynyas in the winter period (Nov.-Apr.) 2007/08, aggregated within the four polynya masks. The grey bars show the histogram of $C_H$ from C05nt. Only values below $6 \cdot 10^{-3}$ have been used for the construction of this figure. The mean values of the C05 simulations are $\approx 2.5 \cdot 10^{-3}$ and the standard deviations are $\approx 0.28 \cdot 10^{-3}$, except for C05-50/5 and C05-50/1 where the mean values are $\approx 2.27 \cdot 10^{-3}$ and $\approx 2.31 \cdot 10^{-3}$, and the standard deviations are $\approx 0.18 \cdot 10^{-3}$, respectively. The constant value of $C_H = 3.0 \cdot 10^{-3}$ of Willmes et al. (2011) is marked with an arrow. The mean $C_H$ value derived from MODIS2km (Nov.-Mar.) is $C_H = 2.3 \pm 0.3 \cdot 10^{-3}$.

be used? This question is not easily answered and is still an open issue. A strategy might be a simultaneous application of both, modelling and remote sensing approaches, in order to compensate for weaknesses. This issue directly impedes the decision of an optimal model configuration.

According to our study, the approach of Willmes et al. (2011) constitutes the closest reference because of the same satellite data that were used to derive polynya area at a comparable horizontal resolution. Although the definition of polynyas are different, the assumption of $0.1\,\mathrm{m}$ thin-ice in areas of SIC$\leq 70\,\%$ is similar to the definition of $\leq 0.2\,\mathrm{m}$ as in Willmes et al. (2011). Larger differences evolve from the assumption on $C_H$ (Fig. 12) and the horizontal resolution of the atmospheric data. Given these deviations, the IP based on C05nt, C05wt10, and C05-50/1 are still close to the results of Willmes et al. (2011). Although the use of MODIS, i.e. higher resolved satellite products, results in higher IP estimates, the reason for this is the higher horizontal resolution that causes larger polynya areas and not the representation of subgrid-scale energy fluxes within polynyas in ERA-Interim, which is still too coarse. For thicker ice the $C_H$ values converge to $\leq 1.5 \cdot 10^{-3}$, a value also reported by Schröder et al. (2003).

Given these issues, the decision which TIT should be used with the TA is another degree of freedom and cannot sufficiently be answered from our study. A justified assumption is to rely on MODIS TIT (Fig. 13). The mean derived TIT for the winter





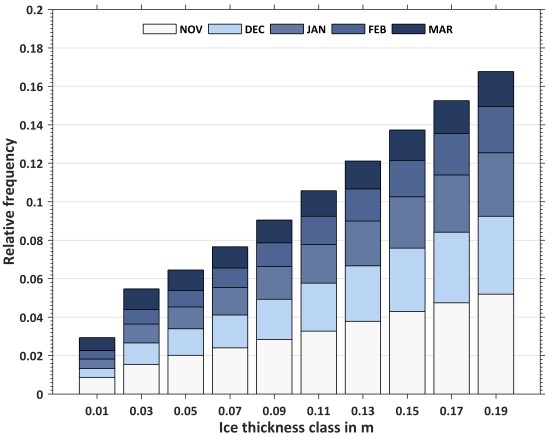

**Figure 13.** Thin-ice thickness distribution ($\leq 20\,\mathrm{cm}$) in the Laptev Sea derived from MODIS data for the winter periods (Nov.-Mar.) 2002/03–2014/15. The bars indicate the relative distribution of each thickness class from the total number of TIT $\leq 0.2\,\mathrm{m}$ appearances between the winter seasons 2002/03 and 2014/15. Contributions of each month with respect to the whole winter season for each thickness class are indicated by the blueish colors (see the legend). The mean thickness ($\pm$ one standard deviation) in this period is $13.5 \pm 0.5\,\mathrm{cm}$ ($8.7\,\mathrm{cm}$ for $\leq 10\,\mathrm{cm}$). In the winter period 2007/08, the mean is $14.0 \pm 2\,\mathrm{cm}$ ($7.7\,\mathrm{cm}$ for $\leq 10\,\mathrm{cm}$).

periods (Nov.-Mar.) 2002/03–2014/15 is $13.5 \pm 0.5\,\mathrm{cm}$, which is slightly thicker than our assumed TIT in CCLM. Unfortunately, the MODIS TIT show no maximum at a specific ice thickness, which gives no preference for the choice of the sub-grid TIT for the tile approach.

    Based on the statistical analysis, two simulations remain with a similar IP to Willmes et al. (2011): C05wt10 and C05-50/1. Although both simulations were performed with a TA, there are two facts that speak for C05-50/1. First, assuming

$50\,\mathrm{cm}$ as a grid-scale ice thickness is realistic because this thickness can be detected by passive microwave sensors, whereas $10\,\mathrm{cm}$ cannot be detected. Given that, it is further realistic to assume a subgrid-scale thickness such as $1\,\mathrm{cm}$, which represents the undetectable ice thickness. Second, the estimations with MODIS indicate a higher daily standard deviation compared to Willmes et al. (2011), which is similar to $\sigma$ of C05-50/1.

## 9    Conclusions

In this study we implemented a tile-approach (TA) for subgrid-scale energy fluxes within fractional sea ice in COSMO-CLM and analysed the sensitivity of sea-ice production (IP) of Laptev Sea polynyas and the effects on the atmospheric boundary layer. The results show that the IP is highly sensitive to the assumptions of ice thickness within polynyas associated with the TA.





The IP within polynyas increased significantly for most simulations if the subgrid-scale ice is thinner than the grid-scale ice. The relative increases were found to range from $+0.3\%$ to $+124.5\%$ due to a stronger heat loss of up to $+109.7\%$ within polynyas.

On one side, the TA improves the physical representation of polynyas in CCLM because fractional sea ice is considered, on the other side a new degree of freedom is introduced as it is unclear which ice thickness should be assumed within polynyas. The derivation of an optimal configuration of CCLM was not intended in this study, and is yet a difficult task because of sparse observed ice thickness distributions within polynyas.

Instead we used remote sensing data as a baseline to compare the simulated IP with. This comparison remained difficult as well because of differences in the definition of polynyas, in the atmospheric forcing and particularly in the calculation of the turbulent diffusion coefficients. The latter were kept either constant for the remote sensing estimations or were calculated from coarse atmospheric data sets, which do not contain polynyas. In CCLM the coefficients were calculated anew every time step with considering polynyas. A next step towards an improved estimation of IP estimations from remote sensing methods could be the use of CCLM data instead of coarse reanalyses. However, Adams et al. (2013) showed that this could also lead to problems for MODIS-based methods because of inconsistencies between the CCLM ice distributions and MODIS surface temperatures.

Nevertheless, based on statistics C05-50/1, which assumes a ice thickness of $50\,\mathrm{cm}$ at grid-scale and $1\,\mathrm{cm}$ at subgrid-scale, simulated the closest ice production with respect to Willmes et al. (2011). Besides the good agreement, this configuration is preferred because the AMSR-E sensor is able to detect $50\,\mathrm{cm}$ thick ice, but not $1\,\mathrm{cm}$ thin ice. Thus, with this model configuration we consider subgrid-scale energy fluxes over fractional sea ice based on reasonable assumptions.

This study shows that CCLM with our implemented TA produces realistic results and improves the representation of polynyas in an atmospheric regional climate model. An extension of our TA would be the separate calculation of the momentum flux for ice and ocean. This would further allow the implementation of a form drag parameterization (Lüpkes et al., 2012; Lüpkes and Gryanik, 2014). The form drag likely increases the turbulence over fractional sea ice and hence the turbulent heat loss over polynyas, which might considerably affect the sea-ice production.

In summary, the implementation of a tile-approach for subgrid-scale energy fluxes within fractional sea ice is a large step forward to adapt COSMO-CLM for applications in polar regions.

## Appendix A: Sea-ice albedo scheme

We implemented a modified Køltzow scheme (Køltzow, 2007) (Fig. 14) to replace the default treatment of sea-ice albedo, which was previously set to $\alpha_i = 0.75$ for ice thickness $> 0.1\,\mathrm{m}$ and $\alpha_i = 0.2$ for ice thickness $\leq 0.1\,\mathrm{m}$ (Schröder et al., 2011). Furthermore, the Køltzow scheme includes a parameterization of melt ponds (see Køltzow (2007) for details), yet they are of no importance for our study. The scheme is based on measurements retrieved during the Surface heat Budget of the Arctic Ocean (SHEBA) project (Uttal et al., 2002). It is forced by the surface temperature $T_{sfc}$, which may be either the ice ($T_i$) or the snow surface temperature ($T_s$) (Fig. 3). If no snow cover is present the albedo only depends on the ice thickness. If the ice




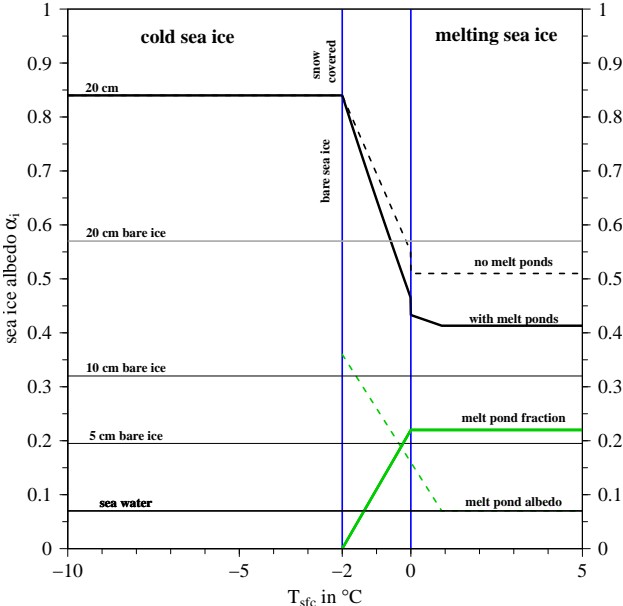

**Figure 14.** Sea-ice albedo resulting from the modified Køltzow-scheme (Køltzow, 2007) in dependence of the ice surface temperature and thickness. Thereby the threshold thickness above which a snow cover of $10\,\mathrm{cm}$ is assumed is $h_c = 0.2\,\mathrm{m}$ (bold black line). In addition the melt pond fraction is shown as a function of the ice temperature (bold green line) and the resulting modification (dashed green line) of the sea-ice albedo (dashed black line). For bare sea-ice (thin black lines) a constant albedo value is assumed, which is linearly decreasing from 0.57 (Persson et al., 2002) at $20\,\mathrm{cm}$ ice thickness (bold grey line) to 0.07 (ocean albedo, Perovich and Grenfell (1981)), but constant over all surface temperatures, as shown in Eq. (A2). The vertical blue lines mark the transition range from cold to melting conditions.

thickness exceeds the threshold value of $h_c = 0.2\,\mathrm{m}$, a snow cover on sea ice is assumed in accordance to the sea-ice module. Sea ice thicker than $h_c$ is treated as thick ice and the albedo is estimated by:

$$
\quad \alpha_i = \begin{cases} 0.84 & \text{if } T_{sfc} \leq -2\,^\circ\mathrm{C} \\ 0.84 - 0.145(2 + T_{sfc}) & \text{if } 0\,^\circ\mathrm{C} > T_{sfc} > -2\,^\circ\mathrm{C} \\ 0.51 & \text{if } T_{sfc} > 0\,^\circ\mathrm{C}. \end{cases} \tag{A1}
$$

Køltzow (2007) sets the albedo for cold sea ice to a high value of 0.84, which is supposed to include the effects of snow on sea ice in winter and spring. In the original scheme Køltzow (2007) set the threshold for thin ice to $h_c = 0.25\,\mathrm{m}$, but since the values above are only valid for snow covered sea ice, we set $h_c = 0.2\,\mathrm{m}$ to be consistent with the sea-ice module.

For thin-ice, we implemented a linear decrease towards the ocean albedo ($\alpha_o = 0.07$):

$$
\quad \alpha_i = \alpha_o + (h_i/h_c) \cdot (\alpha_c - \alpha_o) \tag{A2}
$$

As a starting value we use $\alpha_c = 0.57$, the albedo of thick bare sea ice from Persson et al. (2002).





Fig. 14 shows a summary of both cases. If the ice thickness is at least $0.2\,\mathrm{m}$ (bold black line) then the albedo is constant ($\alpha_i = 0.84$) for cold, snow covered sea ice. It decreases with increasing surface temperature, if $-2\,°\mathrm{C}$ are exceeded. This temperature denotes a threshold where melting begins and sea ice is changing its albedo characteristics. In addition, if melt
ponds occur (black solid line), the albedo is somewhat lower during the melting season. The fraction of melt ponds increases with $T_{sfc} > -2\,°\mathrm{C}$ to a maximum of $22\,\%$ (bold green line), an upper limit set by Køltzow (2007), and the albedo of melt ponds converges to the albedo of sea water (dashed green line). Furthermore, in Fig. 14 the thin-ice albedo is exemplified for four ice thicknesses which are not covered with snow and for which a constant albedo is assumed (thin black lines).

If the tile-approach is used, subgrid-scale open water reduces the grid-average albedo accordingly, compared to a complete
coverage with sea ice. A comparable, though less pronounced, reduction of albedo occurs if 1 cm thin ice coverage is assumed for subgrid-scale open water.

## Appendix B:  Implementation of the tile-approach in CCLM

In order to simulate the subgrid-scale energy fluxes over fractional sea ice, it is necessary to differentiate the energy balance and its components over water and ice. Over sea ice (index $k = i$) or ocean (index $k = o$) the total atmospheric heat flux (see Fig. 3) is:

$$Q_{A,k} = K_k^* + L_k^* + H_k + E_k \tag{B1}$$

with $K_k^*$ the net shortwave radiation, $L_k^*$ the net longwave radiation, $H_k$ the turbulent flux of sensible heat and $E_k$ the turbulent flux of latent heat.
5    All routines of CCLM, except the sea-ice and the turbulence module, calculate with grid-box averaged coefficients or fluxes (flux averaging approach, Vihma (1995)), which is best suited if the sea-ice module only requires the fluxes over ice (Lüpkes and Gryanik, 2014). The procedure is described in section B3.

As initial data the module requires the sea surface temperature (SST), the sea-ice fraction (A) and extent, the sea-ice thickness (SIT), the surface temperature of sea ice ($T_i$), specific humidity at the ice surface, the wind-speed on the lowest model level,
10    and incoming longwave and shortwave radiation (see Schröder et al. (2011) for more details).

The calculation of the components of the energy balance equations are shown in the next subsections.

### B1    Shortwave radiation

The grid-box average of the albedo $\alpha_m$ (index $m$ for 'mixed') is calculated as:

$$\alpha_m = A \cdot \alpha_i + (1 - A) \cdot \alpha_o, \tag{B2}$$

with $A$ the sea-ice fraction, $\alpha_o = 0.07$ the albedo of the ocean, and $\alpha_i = f(T_i, h)$ the albedo of sea ice as a function of sea ice
temperature ($T_i$) and thickness ($h$) (see section A). Based on this mixed albedo the upward shortwave radiation is calculated





as:

$$K \uparrow_m = \alpha_m \cdot K \downarrow, \tag{B3}$$

with $K \downarrow$ the incoming shortwave radiation. The grid-box average net shortwave radiation is calculated as:

$$K_m^* = K \downarrow - K \uparrow_m = (1 - \alpha_m) \cdot K \downarrow. \tag{B4}$$

This grid-box averaged net shortwave radiation is the input for the sea-ice module where the upward shortwave radiation over ice $K \uparrow_i$ is calculated as:

$$K \uparrow_i = \alpha_i \cdot K \downarrow, \tag{B5}$$

The final net shortwave radiation over sea ice or ocean becomes:

$$K_k^* = (1 - \alpha_k) \cdot K \downarrow = \frac{1 - \alpha_k}{1 - \alpha_m} \cdot K_m^* \tag{B6}$$

where the index $k$ refers either to $i$ (sea ice) or $o$ (ocean).

## B2 Longwave radiation

The subgrid-scale ocean surface temperature ($T_o$) is assumed to be at the freezing point ($-1.7\,^{\circ}\mathrm{C}$) if open water is assumed, or to be a prognostic variable if a thin-ice cover is assumed. The ice surface temperature ($T_i$) is also a prognostic variable in the sea-ice module.

To account for subgrid-scale longwave radiation, we calculate the upward longwave radiation over sea ice and ocean as:

$$L \uparrow_k = \epsilon \sigma T_k^4 - (1 - \epsilon) L \downarrow, \tag{B7}$$

with $\sigma$ the Stefan–Boltzmann constant, $L \downarrow$ the incoming longwave radiation, $\epsilon$ the surface emissivities of sea water and ice, which are assumed to be equal ($\epsilon = 0.996$), and $T_k$ the surface temperature of ice or ocean.

Then the net longwave radiation balance over sea ice or ocean becomes:

$$L_k^* = L \downarrow - L \uparrow_k. \tag{B8}$$

## B3 Turbulent fluxes of sensible and latent heat

We modified the parameterization of the turbulent fluxes of sensible ($H$) and latent heat ($E$) within a grid box, in contrast to the standard version of CCLM and the sea-ice module of Schröder et al. (2011). Over sea ice or ocean the roughness length $z_0$ and the turbulent coefficients of heat and moisture $C_H$ were previously calculated from the predominant surface type of a grid

5 box: ice or sea water. We modified this procedure by a tile-approach; now the fluxes are calculated both for sea ice and ocean within a grid box with different $z_0$ and $C_H$. Afterwards they are averaged in a 'flux-averaging approach' and an average $C_H$





is calculated for other modules. The calculation of the momentum flux is not modified and for the details of the calculation we refer the reader to Doms et al. (2011).

In CCLM a stability and roughness length dependent surface flux formulation is used, which is based on flux calculations after Louis (1979). The fluxes are calculated with a bulk approach:

$$H = -\rho c_p C_H |v_h| (\Theta_{sfc} - \Theta) \tag{B9}$$

$$E = -\rho L_f C_H |v_h| (q_{sfc} - q) \tag{B10}$$

with $\rho$ the air-density, $c_p$ the heat capacity of air, $\Theta$ and $\Theta_{sfc}$ the potential temperature at the lowest model layer and at the surface (ice or ocean). $q$ and $q_{sfc}$ are the specific humidity at the lowest model layer and at the surface (ice or ocean), $L_f$ the latent heat of fusion (and sublimation in case of sea ice), $|v_h| = \sqrt{u^2 + v^2}$ the absolute wind speed, and $C_H$ the turbulent transfer coefficient for heat and moisture.

To calculate the turbulent transfer coefficients it is first necessary to calculate the roughness length of sea-water ($z_{0,o}$) and sea ice ($z_{0,i}$). In case of sea ice we set $z_{0,i} = 0.001\,\mathrm{m}$ as in Schröder et al. (2011). Over open water a modified Charnock-formula is used (see Doms et al., 2011). In case of $H$ and $E$, we assume the additional roughness length for heat $z_h$ (Doms et al., 2011) to be equal to $z_0$ over subgrid-scale open ocean within the sea-ice cover.

The transfer coefficients are calculated over sea ice ($C_{H,i}$) and ocean ($C_{H,o}$), respectively. The turbulent fluxes over sea ice ($H_i$, $E_i$) and ocean ($H_o$, $E_o$) can be retrieved by inserting these coefficients into Eqs. (B9-B10). Then all terms of Eq. (B1) are known to solve the energy balance over both surface types.

The fluxes of sensible and latent heat, the turbulent transfer coefficient for heat, and the surface temperature are averaged according to the sea-ice concentration $A$:

$$H_m = A \cdot H_i + (1 - A) \cdot H_o \tag{B11}$$

$$E_m = A \cdot E_i + (1 - A) \cdot E_o, \tag{B12}$$

$$C_{Hm} = A \cdot C_{H,i} + (1 - A) \cdot C_{H,o}, \tag{B13}$$

$$T_{sfc} = A \cdot T_i + (1 - A) \cdot T_o \tag{B14}$$

The grid averaged temperature fields are used for the comparisons in section 6.

*Author contributions.* O. Gutjahr implemented the tile-approach and other modifications to the CCLM source code, conducted the CCLM simulations, designed the study and wrote the paper. G. Heinemann assisted in designing the experiments and the structure of the paper. A. Preußer wrote the section on MODIS and calculated the sea-ice production from MODIS data, S. Willmes calculated the ice production rates based on AMSR-E and NCEP and assisted in the discussion on the comparison of remote sensing and modelling results. C. Drüe contributed to the translation of the equations into source code.





*Acknowledgements.* This work was funded by the German Ministry for Education and Research under grant 03G0833D, and is part of the German-Russian Transdrift project. We thank the CLM-Community and the German Weather Service for providing the basic COSMO-CLM model. The AMSR-E data was provided by the University of Bremen, the MODIS data were provided by the US National Snow and Ice Data Center, ERA-Interim by the ECMWF, and the PIOMAS data set by the Polar Science Center (University of Washington). Finally, we thank the DKRZ for providing the computational time.



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
