# Peer review of "Quantification of ice production in Laptev Sea polynyas and its sensitivity to thin-ice parameterizations in a regional climate model"

_The Cryosphere, 2016_

## Referee Comment (RC1) · Anonymous Referee #1 · 28 Jun 2016

This paper describes a range of different set-ups for a high resolution atmospheric model simulating polynyas in the Laptev Sea. The set-up involves the use of "tiles" which are sub-grid scale parametrization of thin sea ice. The quality of the model appears fine, and the number of simulations are impressive. Citations given are generally fine, and the technical quality of the figures are OK. Polynyas are important regions with substantial ice production and very high heat fluxes, and are thus an important part of the Cryosphere.

With the above in place one would expect that the paper would be in fairly good shape,

but I am sorry to state that this is not the case. The attention to detail is totally over-whelming, and there is almost nothing learned in terms of physics. The use of abbrevi-ations also has no end, and is a clear sign that at this level the text is more like a data report intended for those that may apply the same model system in a future study. One look at Table 3 should say it all.

For the main conclusion not much has been learned about the atmospheric boundary layer, where the model actually should resolve processes in a better way than earlier model attempts. That the overall heat loss increases along with the total ice production when areas of thinner ice is added as a new lower boundary condition is indeed what is expected and does not contribute to an increased scientific understanding.

The text is also written only from a modellers perspective, without even the most basic understanding of processes in a polynya in nature. Moving downwind from the beach there is CONTINOUS change from open water to thicker and thicker sea ice, much like the MODIS observations plotted in Figure 13. In essence resolving the heat fluxes and the ice thickness inside a polynya is a coupled problem. Such coupling has been done albeit in a very simple way starting with Pease (1987). I'm not saying that you should invent a new downwind thickness parametrization for thickness, but rather state that you have made your choices, and then how this is simplified from nature.

A statement like "open water areas particularly produce new ice and are hence rarely free of ice" (Page 6, line 10) is not correct at all. Polynyas stay open for many hours during strong wind events that effectively transport sea ice (frazil, grease, pancake, solid ice) downwind (Morales-Maqueda et al 2004, Fig. 17). An open polynya length along the wind direction between 10 – 30 km is not uncommon.

The most interesting part of this study is the response of the atmospheric boundary layer, as shown in Figure 7. But here two plots should be shown, the "best" case and the similar without the tiles (C05nt – perhaps, it is just a total misuse of abbreviations here). This should be the case also for Figure 5, Figure 6, and Figure 11. All the tables

should only compare values between your "best" model simulation and the one without the tiles. The details are not interesting, unless you have some way of evaluating the model performance.

The paper needs to be totally rewritten if it is to be published as a scientific article. First – make your choice on the "best" model simulations, and present all relevant results to this one case first. Then compare to existing simulations without the tiles. At the end you can include some sensitivities to some of the different choices made, like the different thin ice thickness' inside the polynya.

This reviewer has not been convinced that new scientific understanding has been achieved here, but I'm willing to review a new version of a totally rewritten paper if that is submitted.

Sorry to be so negative, but this version can be saved as a technical report for researchers that will work on the same model in the future. No one else would have the interest to read about all these details, and you have not done the important scientific job it is to extract the new understanding based on your model simulations.

. . .............

New Citations:

Pease, C.H. 1987. The size of wind-driven coastal polynyas. J. Geophys. Res., 92(C7), 7049–7059.

---

## Referee Comment (RC2) · Anonymous Referee #2 · 1 Jul 2016

General comments:

The authors implement a tile-approach into the atmospheric COSMO-CLM model to account for subgrid-scale sea-ice inhomogeneities and examine the impact on estimated sea ice production in the Laptev Sea polynyas. Due to huge differences between sea-ice and open water properties and the linear dependency of energy fluxes on these properties, the implementation of the tile-approach is a significant improvement of the COSMO-CLM model. The configuration of the simulations is complex: 182 daily simulations for each setup with initial conditions based on AMRS-E sea ice

concentration, PIOMAS sea ice thickness, and MODIS based assumption for polynya sea-ice thickness and sub-grid scale ice thickness. Overall the setup is convincing and promises to give the most realistic results for ice production given the limitations of the very simple sea ice module in COSMO-CLM. In addition to the main finding that the ice production can increase by a factor of 2, the authors discuss a number factors documenting how difficult it is to accurately determine ice production in polynyas.

In spite of the quality of the simulations and the significance of the results major revisions are required. My main concern is that the current manuscript fails to appeal to scientists who are not familiar with polynya processes in the Laptev Sea and/or are not using the COSMO-CLM model. Required background information and motivation (e.g. formation of polynyas, importance of polynya ice production) are missing in the study. The discussion of the results is too restricted to differences with one remote sensing product (Willmes et al., 2011) and potential adjustments of the COSMO-CLM model. What can somebody learn from this study who is not using COSMO-CLM? Prescribing the subgrid-scale ice thickness cannot be the best solution to simulate polynya processes. What are consequence from your study for applying a more complex sea ice model which aims to simulate the processes?

Specific comments:

1. Abstract: too long; too many details about setup; mention that COSMO-CLM is atmospheric model; not clear whether numbers are winter averages or extremes from case studies. Better give numbers from preferred reference run. Last sentence too specific for COSMO-CLM setup (see General Comments.)

2. Introduction: mainly just technical introduction; paragraph about polynyas and their importance missing; mention discrepancy of estimates of polynya ice production from previous studies.

3. Configuration of CCLM / initial conditions: If a model grid point has e.g. SIC = 60 %, is this grid box regarded as polynya box in which 60% of the area is covered by

ice with a thickness of TIT (10 cm for model run C05wt1) and 40% with a thickness of the subgrid-scale TIT (1 cm for this model run)? Do the ice thickness, TIT and subgrid-scale TIT change during the 24h simulation period?

4. Page 8, line 13: "the turbulent exchange coefficient CH is variable in time": Why? Please write "is a function of ..."

5. Verification with in situ data: The AWS were deployed over the fast ice and Table 3 and Figure 4 document that there are no significant differences between the sensitivity runs as long SIC > 95 %. Just show results from one simulation (CO5nt) in Table 3. Figure 4 could be omitted in my opinion.

6. Case study on 4 January 2008: The differences in Figure 5 and 6 are quite difficult to spot and at this stage the reader is not aware whether you have a preferred reference run. Might be helpful to change order and to focus just on the preferred run for the case study.

7. Conclusions: Only present numbers from your preferred reference run. Put your results in wider context. See general comment. Add a paragraph about how your results might help to simulate polynya processes using a more complex sea ice model including recent advances of frazil ice modules (e.g. Wilchinsky et al., JPO 2015).

8. Give numbers with adequate decimals in text and tables (e.g. +110% instead of 109.7% in line 11 or 29 km3 instead of 29.05 km3 in line 14).

---

## Author Comment (AC2) · 29 Aug 2016

Appendix to our previous post:

In wintertime, only a very small percentage of a polynya is ice-free, as it was shown by many studies and is illustrated by the following two figures from satellite data and airborne in-situ observations (from Willmes et al. 2010).

[Figure]

[Figure]

**Figure 2.** The central part of the figure is a SAR image of a part of the WNS polynya surveyed by helicopter during the polynya event on 30 April 2008 (compare **Figures 1** and **3a**). Sections I–V indicate zones of different ice conditions (compare **Figure 8**) with locations of aerial photographs (numbers 1–8) across the polynya. Photographs were taken at a height of 50 m, covering a footprint of 60 m × 40 m. The HEM-Bird is visible in the centre of images 1–8.

**Fig. 1.** Ice-free polynyas in the Laptev Sea (from Willmes et al. 2010).

[Figure]

---

## Author Response (AR1)

This paper describes a range of different set-ups for a high resolution atmospheric model simulating polynyas in the Laptev Sea. The set-up involves the use of "tiles" which are sub-grid scale parametrization of thin sea ice. The quality of the model appears fine, and the number of simulations are impressive. Citations given are generally fine, and the technical quality of the figures are OK. Polynyas are important regions with substantial ice production and very high heat fluxes, and are thus an important part of the Cryosphere.
With the above in place one would expect that the paper would be in fairly good shape, but I am sorry to state that this is not the case.

R#1: The attention to detail is totally overwhelming, and there is almost nothing learned in terms of physics.
A: We agree that many details of the results were included in the manuscript, which could make it difficult to focus on the main aspects. The primary objective of the paper was to assess the sensitivity of ice production of Laptev Sea polynyas on the chosen assumption for thin-ice thickness of a tile approach for subgrid-scale energy fluxes. This is not a specific problem of the used model (CCLM), but a general problem of all regional climate models using the tile approach. To our knowledge, it is generally assumed that the subgrid fraction not covered by sea ice is assumed to be open water (e.g. in the recently published ASR data set). We could show that the ice production is very sensitive to the tile-approach and thin-ice thickness, which affects also the atmospheric boundary layer structure. However, we agree that for example the latter issue was not discussed sufficiently.
Changes in the manuscript: In the revised manuscript we change the structure of the paper and focus more on the physical aspects by simultaneously reducing the details of the results, i.e. we present only the results of three simulations and show sensitivities only where useful. We further changed the title to: "Quantification of ice production in Laptev Sea polynyas and its sensitivity to thin-ice parameterizations in a regional climate model" to better reflect to content of the manuscript.

R#1: The use of abbreviations also has no end, and is a clear sign that at this level the text is more like a data report intended for those that may apply the same model system in a future study. One look at Table 3 should say it all.
A: Multi-model or sensitivity studies always include a lot of abbreviations. We accept this remark and thus reduce the amount of details and abbreviations to a necessary minimum.
Changes in the manuscript: The abbreviations of the simulation runs will be homogenized and we will change the structure of the manuscript so that it focuses on the scientific aspects not on the technical details. Therefore, we restrict the presented results to three simulations: C05nt (the reference), C05wt0 (subgrid-scale open-water scenario) and C05-50/1 (most realistic assumptions). We will change these abbreviations to: C05-ref, C05-10/0, and C05-50/1. Table 3 will be condensed.

R#1: For the main conclusion not much has been learned about the atmospheric boundary layer, where the model actually should resolve processes in a better way than earlier model attempts. That the overall heat loss increases along with the total ice production when areas of thinner ice is added as a new lower

boundary condition is indeed what is expected and does not contribute to an increased scientific understanding.

A: We agree that we did could include more results on the ABL. However, we have already addressed some important aspects (impact on the warm plume formation, turbulence structure, cloud formation), which contribute to an increased quantitative understanding of the processes and their feedbacks.

Changes to the manuscript: We will rewrite the ABL part to point out the main conclusions.

R#1: The text is also written only from a modellers perspective, without even the most basic understanding of processes in a polynya in nature. Moving downwind from the beach there is CONTINOUS change from open water to thicker and thicker sea ice, much like the MODIS observations plotted in Figure 13. In essence resolving the heat fluxes and the ice thickness inside a polynya is a coupled problem. Such coupling has been done albeit in a very simple way starting with Pease (1987). I'm not saying that you should invent a new downwind thickness parametrization for thickness, but rather state that you have made your choices, and then how this is simplified from nature.

A: We see the point that the text is focused too much on the modeller's perspective, however it is not clear to us how we missed "the most basic understanding" of polynya processes. We did not intent to give a too detailed introduction on polynya processes and thus cited relevant papers for more information. But we agree that some more information on e.g. polynya formation and the spatial structure of thin-ice within a polynya are useful additions. We are aware that the ice thickness increases with downwind direction, which is not represented in CCLM yet. Figure 13 shows the spatio-temporal histogram of thin-ice within Laptev Sea polynyas retrieved from MODIS data, which is not to confuse with the spatial sequence of thin-ice in a polynya. Our implementations to CCLM are just the first step to represent fractional sea ice, which was not present in all CCLM simulations before. In this context, we would like to note that even Polar-WRF does not use spatial distributions of thin-ice within polynyas, in fact in WRF there is always subgrid-scale open-water assumed, which is much more unrealistic then our assumptions. Comparing e.g. Fig.11a and Fig.11c there is still a downwind structure of the ice production visible for the WNS polynya (opened on 30 April 2008), which is not present in the reference simulation (Fig.11a) (and weaker for the other simulation runs).

Changes in the manuscript: We will comment on our chosen assumptions on the thin-ice distribution and that it is a simplification to the thin-ice structure observed in nature. This is an important point we missed to mention in the manuscript.

R#1: A statement like "open water areas particularly produce new ice and are hence rarely free of ice" (Page 6, line 10) is not correct at all. Polynyas stay open for many hours during strong wind events that effectively transport sea ice (frazil, grease, pancake, solid ice) downwind (Morales-Maqueda et al 2004, Fig. 17). An open polynya length along the wind direction between 10 – 30 km is not uncommon.

A: We guess our formulation might be too imprecise as we actually meant that the heat loss is highest over open-water areas. These open-water areas quickly produce frazil and grease ice, which is then advected downstream and consolidates to thicker ice, hence the continuous increase in thickness mentioned in the previous comment. However, based on field experience of the authors we argue that the fraction of the Laptev polynya area that is completely free of ice is

relatively small during winter (as illustrated in the appendix and Fig.13 of our paper).
Changes in the manuscript: We will reformulate this sentence to make clear what we wanted to express and we further add the information that in our simulations it is assumed that new ice is instantly advected downstream so that the initial thin-ice thickness is restored after every time step. We will also add the word „wintertime" to the polynyas.

R#1: The most interesting part of this study is the response of the atmospheric boundary layer, as shown in Figure 7. But here two plots should be shown, the "best" case and the similar without the tiles (C05nt – perhaps, it is just a total misuse of abbreviations here). This should be the case also for Figure 5, Figure 6, and Figure 11. All the tables should only compare values between your "best" model simulation and the one without the tiles. The details are not interesting, unless you have some way of evaluating the model performance.
A: We think that the most interesting part is the ice production, since this has impacts also for the ocean circulation. We will pick up the suggestion of taking the "best" model simulation as reference.
Changes in the manuscript: As mentioned above we will restrict the presentation of results to three simulation runs. That is we reduce the amount of subplots of the mentioned figures and also reduce the tables to a necessary minimum.

R#1: The paper needs to be totally rewritten if it is to be published as a scientific article. First – make your choice on the "best" model simulations, and present all relevant results to this one case first. Then compare to existing simulations without the tiles. At the end you can include some sensitivities to some of the different choices made, like the different thin ice thickness' inside the polynya. This reviewer has not been convinced that new scientific understanding has been achieved here, but I'm willing to review a new version of a totally rewritten paper if that is submitted.
Sorry to be so negative, but this version can be saved as a technical report for researchers
that will work on the same model in the future. No one else would have the interest to read about all these details, and you have not done the important scientific job it is to extract the new understanding based on your model simulations.

A: We have a different opinion concerning the reviewer's statements about scientific understanding and technical report, but we will restructure and rewrite the manuscript also considering the remarks of reviewer #2 (who states that we show the significant results).
Changes in the manuscript: As mentioned above we will present and compare the results of three simulation runs: a reference without the tile-approach, one run with subgrid-scale open-water as a possible upper limit, and one run which we think is the most realistic configuration.

**Appendix**

**Ice-free polynyas in the Laptev Sea**

In wintertime, only a very small percentage of a polynya is ice-free, as it was shown by many studies and is illustrated by the following two figures from satellite data and airborne in-situ observations (from Willmes et al. 2010).

[Figure]

**Figure 2.** The central part of the figure is a SAR image of a part of the WNS polynya surveyed by helicopter during the polynya event on 30 April 2008 (compare **Figures 1** and **3a**). Sections I–V indicate zones of different ice conditions (compare **Figure 8**) with locations of aerial photographs (numbers 1–8) across the polynya. Photographs were taken at a height of 50 m, covering a footprint of 60 m × 40 m. The HEM-Bird is visible in the centre of images 1–8.

[Figure]

**Figure 3.** Western New Siberian polynya. (a) Envisat SAR backscatter for 30 April 2008 at 0237 UTC. Characteristic backscatter bands B1–B3 and the helicopter flight track at 0225 UTC are indicated. (b) Surface temperature (between $-14\,°C$ and $-4\,°C$) as derived from AVHRR IR brightness temperatures from 29 April 2008 at 2000 UTC. (c) Composite of (a) and (b) together with contour lines (0.05, 0.1, 0.2, and 0.5 m) of the thermal ice thickness $hi_{TH}$ (derived from data in (b)).

An additional example is shown by Adams et al. (2013):

[Figure]

Fig. 1. (a) Map of MODIS (MOD29) ice-surface temperatures in the Laptev Sea on 26 March 2009 1220 UTC. The black line broadly indicates the polynya region in the Laptev Sea. The mask is based on [37]. The small rectangle in (a) locates the subset shown in (b). The north-east oriented stripes around $-15\,°C$ located at $120°-140°$ E and $75°-78°$ N are clouds not identified by the MODIS cloud mask. (b) Subset of the Laptev Sea. Triangles in the map denote the positions of five aerial photographs which were taken on 26 March 2009 0800 UTC during the TRANSDRIFT XV expedition. Black pixels (shown in gray in color version of this figure) in map (a) and (b) mark data gaps due to clouds.

**Anonymous Referee #2**

General comments:
The authors implement a tile-approach into the atmospheric COSMO-CLM model to account for subgrid-scale sea-ice inhomogeneities and examine the impact on estimated sea ice production in the Laptev Sea polynyas. Due to huge differences between sea-ice and open water properties and the linear dependency of energy fluxes on these properties, the implementation of the tile-approach is a significant improvement of the COSMO-CLM model. The configuration of the simulations is complex: 182 daily simulations for each setup with initial conditions based on AMRS-E sea ice concentration, PIOMAS sea ice thickness, and MODIS based assumption for polynya sea-ice thickness and sub-grid scale ice thickness. Overall the setup is convincing and promises to give the most realistic results for ice production given the limitations of the very simple sea ice module in COSMO-CLM. In addition to the main finding that the ice production can increase by a factor of 2, the authors discuss a number factors documenting how difficult it is to accurately determine ice production in polynyas.
In spite of the quality of the simulations and the significance of the results major revisions are required.

R#2: My main concern is that the current manuscript fails to appeal to scientists who are not familiar with polynya processes in the Laptev Sea and/or are not using the COSMO-CLM model. Required background information and motivation (e.g. formation of polynyas, importance of polynya ice production) are missing in the study.
A: Although we mentioned some aspects of polynya formation in the Laptev Sea, we agree that too few information is given for readers which are not familiar with polynyas and how they are implemented in regional climate models besides CCLM.
Changes in the manuscript: We will add more information and details on the polynya processes in the Laptev Sea and how polynyas are represented in RCMs. Further we will state more clearly the objectives of our study.

R#2: The discussion of the results is too restricted to differences with one remote sensing product (Willmes et al., 2011) and potential adjustments of the COSMO-CLM model. What can somebody learn from this study who is not using COSMO-CLM? Prescribing the subgrid-scale ice thickness cannot be the best solution to simulate polynya processes. What are consequence from your study for applying a more complex sea ice model which aims to simulate the processes?
A: We chose to compare our results to the estimates of Willmes et al. (2011) because it is based on the same polynya masks and the same satellite date (i.e. on the same original AMSR-E product). We think with the product of Willmes et al. (2011) we chose the most suited product available for our comparison, as mentioned in the manuscript. Otherwise, even more issues arise for comparisons with model results.
We think that the results of the sensitivity study are valuable also for other models using the tile approach and prescribed sea ice coverage. Although we use a rather simple approach to represent subgrid-scale ice thickness, some of the issues remain even if more complex approaches are used. Subgrid open water or thin ice fraction is also a problem for complex sea-ice/ocean models.
Changes in the manuscript: We will adapt the discussion section by commenting on the general relevance of our results for other RCMs and consequences if more complex sea ice models are used within an RCM. As far as the remote sensing

product is concerned, we already tried to generalize from our results, so we do not see the requirement to adapt the paragraphs dealing with this issue.

R#2:Specific comments:
1. Abstract: too long; too many details about setup; mention that COSMO-CLM is atmospheric model; not clear whether numbers are winter averages or extremes from case studies. Better give numbers from preferred reference run. Last sentence too specific for COSMO-CLM setup (see General Comments.)
A: We agree on the issues raised by the reviewer.
Changes in the manuscript: The abstract will be revised and shortened considerably. We will also make clear what the numbers represent.
2. Introduction: mainly just technical introduction; paragraph about polynyas and their importance missing; mention discrepancy of estimates of polynya ice production from previous studies.
A: We put the paragraph on polynyas within section 2 as 2.1. This was not the best option and the introduction (but also the general structure of the manuscript) needs to be overdone.
Changes in the manuscript: We will move and integrate section 2.1 to the introduction and add more background information on polynyas, their importance and what was not represented in previous studies.
3. Configuration of CCLM / initial conditions: If a model grid point has e.g. SIC = 60%, is this grid box regarded as polynya box in which 60% of the area is covered by ice with a thickness of TIT (10 cm for model run C05wt1) and 40% with a thickness of the subgrid-scale TIT (1cm for this model run)? Do the ice thickness, TIT and subgrid-scale TIT change during the 24h simulation period?
A: This is correct, the SIC of AMSRE constitutes the grid-scale ice thickness (10cm) in this example and 1-SIC is the 'open-water' fraction or the area with subgrid-scale ice thickness. However, this differentiation is not restricted to polynyas but is applied generally for fractional sea ice. The ice thickness is allowed to change within a time step, but is restored after every time step.
Changes in the manuscript: We will reformulate the sentences concerning these two issues to make the procedure more clearly.
4. Page 8, line 13: "the turbulent exchange coefficient CH is variable in time": Why? Please write "is a function of …"
Changes in the manuscript: We will reformulate the sentence.
5. Verification with in situ data: The AWS were deployed over the fast ice and Table 3 and Figure 4 document that there are no significant differences between the sensitivity runs as long SIC > 95 %. Just show results from one simulation (CO5nt) in Table 3. Figure 4 could be omitted in my opinion.
A: We agree on this comment.
Changes in the manuscript: We will remove Fig.4 from the manuscript and restrict Tab. 3 to three simulations (C05nt, C05wt0, and C05-50/1). We will also change the abbreviations of the simulation runs to: C05, C05-10/1, C05-50/1 for a better readability.
6. Case study on 4 January 2008: The differences in Figure 5 and 6 are quite difficult to spot and at this stage the reader is not aware whether you have a preferred reference run. Might be helpful to change order and to focus just on the preferred run for the case study
A: We agree on this comment.
Changes in the manuscript: We will restrict Fig.5 and Fig.6 to only three simulation runs (see comment before). Further we will introduce in the beginning of section 2, what configuration is the reference and what is the optimal one in our opinion so that the reader can follow our chain of arguments more easily.
7. Conclusions: Only present numbers from your preferred reference run. Put your results in wider context. See general comment. Add a paragraph about how your

results might help to simulate polynya processes using a more complex sea ice model including recent advances of frazil ice modules (e.g. Wilchinsky et al., JPO 2015).

A: We agree on this comment.

Changes in the manuscript: We will extend the conclusion section by mentioning what our results mean for more complex sea ice models and restrict the presentation of numbers to our preferred simulation.

8. Give numbers with adequate decimals in text and tables (e.g. +110% instead of 109.7% in line 11 or 29 km3 instead of 29.05 km3 in line 14).

Changes in the manuscript: We will change the numbers accordingly to improve the readability of the manuscript.

[revised manuscript text omitted]

AWS1

C05nt C05-ref

C05wt10 -20.60 1.91 0.79 0.48 3.42 1.56 0.75 0.74 -30.18 26.11 0.80 < 0.01* C05wt1 -20.54 1.92 0.79 0.66 3.46 1.58 0.75 0.55 -30.05 26.25 0.80 < 0.01* C05wt0 C05-10/0

C05-50/5 -20.58 1.91 0.79 0.55 3.43 1.57 0.75 0.68 -30.12 26.14 0.80 < 0.01* C05-50/1

AWS2

C05nt C05-ref

C05wt10 -20.54 2.15 0.78 < 0.01* 3.34 1.33 0.69 < 0.01* -28.36 24.97 0.75 < 0.01* C05wt1 -20.50 2.12 0.79 < 0.01* 3.35 1.35 0.69 < 0.01* -28.72 25.15 0.75 < 0.01* C05wt0 C05-10/

C05-50/5 -20.52 2.14 0.79 < 0.01* 3.35 1.34 0.69 < 0.01* -28.43 24.92 0.76 < 0.01* C05-50/1

AWS3

C05nt C05-ref

C05wt10 -19.53 3.06 0.86 0.09 3.16 1.46 0.70 < 0.01* -23.54 28.65 0.70 < 0.01* C05wt1 -19.47 3.08 0.86 0.13 3.17 1.46 0.69 < 0.01* -23.52 28.76 0.70 < 0.01* C05wt0 C05-10/0

C05-50/5 -19.50 3.08 0.86 0.11 3.16 1.46 0.70 < 0.01* -23.39 28.74 0.70 < 0.01* C05-50/1

AWS4

C05nt C05-ref

C05wt10 -16.31 3.28 0.65 < 0.01* 4.64 2.28 0.92 0.14 -34.31 28.26 0.73 < 0.01* C05wt1 -16.25 3.26 0.66 < 0.01* 4.72 2.38 0.91 0.09 -34.21 28.46 0.74 < 0.01* C05wt0 C05-10/0

C05-50/5 -16.26 3.29 0.65 < 0.01* 4.65 2.29 0.92 0.13 -34.24 28.43 0.73 < 0.01* C05-50/1

water within a polynya is at the freezing point, all energy loss to the atmosphere through the ocean surface is compensated by

freezing. Hence sea-ice growth only occurs if the total atmospheric energy flux over ice (index $k = i$) or ocean (index $k = o$) $Q_{A,k} = K_k^* + L_k^* + H_k + E_k$ is negative, i.e. the ocean looses heat:

$$\frac{\partial h_i}{\partial t} = -\frac{Q_{A,k}}{\rho_i \cdot L_f}, \tag{1}$$

with $h_i$ the sea-ice thickness, $\rho_i = 910\,\text{kg m}^{-3}$ the density of sea ice and $L_f = 0.334 \times 10^6\,\text{J kg}^{-1}$ the latent heat of fusion. We restricted this estimation to the four polynya areas in the Laptev Sea (see Fig. 2), which are identical to those of Willmes et al. (2011). Hence, direct comparisons of our results with estimations from remote sensing  were possible.

We further calculated the IP using the MOD/MYD29 sea-ice surface temperature product (Hall et al., 2004; Riggs et al., 2006) derived from MODIS Terra and Aqua data. In combination with ERA-Interim data ( 2 m temperature, 2 m dew point temperature, 10 m horizontal wind components and pressure at mean sea level), an energy balance model (e.g. Yu and Lindsay, 2003; Adams et al., 2013; Preußer et al., 2015b, a) was applied to derive thin-ice thicknesses up to $0.2\,\text{m}$ at a horizontal resolution of about $2\,\text{km}$. We refer to this estimation as MODIS2km. The turbulent fluxes of sensible and latent heat were calculated by an iterative bulk approach (Launiainen and Vihma, 1990) based on the Monin-Obukhov similarity theory. Thereby, the turbulent exchange coefficient $C_H$ is  a function of stability, and of the roughness length for momentum and for heat, respectively (Doms et al., 2011). Shortwave radiation is not considered as the method is restricted to  nighttime conditions during winter. This method is only applicable to clear sky conditions, as clouds and fog impede an estimation of sea-ice surface temperature (Riggs et al., 2006). Therefore the number of useful swaths per day is variable. For instance, in the Laptev Sea there are about 10 to 14 swaths per day (2002/03 to 2014/15 (Nov.-Mar.)).

Cloud-induced gaps in our daily sea-ice surface temperature and thin-ice thickness composites were filled by a spatial feature reconstruction procedure (Paul et al., 2015; Preußer et al., 2015a). This method interpolates information of previous and subsequent days to fill gaps caused by cloud-cover. Based on these corrected composites and using the method described in Preußer et al. (2015b), ice production rates were calculated for each pixel with an ice thickness $\leq 0.2\,\text{m}$, i.e. for polynya areas.

In a sensitivity analysis of this method (without the spatial feature reconstruction), Adams et al. (2013) stated an uncertainty for the ice-thickness retrieval of $\pm 1.0\,\text{cm}$, $\pm 2.1\,\text{cm}$ and $\pm 5.3\,\text{cm}$ for thin-ice classes of $0 - 5\,\text{cm}$, $5 - 10\,\text{cm}$ and $10 - 20\,\text{
[revised manuscript text omitted]
 $< -1000$ W m$^{-2}$ (Fig.7c), in C05wt1 with $\approx -1000$ W m$^{-2}$ (Fig.7b), in C05-50/5 with $\approx -750$ W m$^{-2}$ (Fig.7e), and $-750$ W m$^{-2}$ to $-1000$ W m$^{-2}$ in C05-50/1 (Fig.7f). Thus if the TA is used with our assumed ice thicknesses, more heat is released into the atmospheric boundary layer.~~

[Figure]

**Figure 7.** Temporal means of the energy balance $Q_A$ and its components averaged over polynya grid boxes for the winter period 2007/08. For the averaging at least 9 grid boxes in the model domain had to be polynyas to include them in the calculation.

−2.5 % −3.1 % −2.9 % +0.6 % +0.9 % −3.0 % −6.5 % C05wt1 −414.4 −267.3 −91.6 −82.6 27.1 −9.8 −10.0 % $Q_A$ -64.5 % 22.1 % 19.9 % −6.5 % - - Δ +64.1 % +60.3 % +123.4 % +22.6 % +19.9 % +46.3 % +61.3 % C05wt0 −529.0 −324.8 −141.3 −90.6 27.7 −11.6 −10.4 % $Q_A$ -61.4 % 26.7 % 17.1 % −5.2 % - - Δ +109.7 % +94.8 % +244.6 % +34.4 % +22.6 % +73.1 % +67.7 % C05-50/5 −187.7 −107.7 −36.0 −61.1 17.1 −4.1 −0.3 % $Q_A$ -57.4 % 19.2 % 32.6 % −9.1 % - - Δ −25.9 % −35.4 % −12.2 % −9.3 % −24.3 % −38.8 % −95.2 % C05-50/1 −303.2 −178.6 −73.0 −70.6 19.0 −6.5 −0.7 % $Q_A$ - 58.9 % 24.1 % 23.3 % −15.9 % - - Δ +20.1 % +7.1 % +78.0 % +4.7 % −6.3 % −3.0 % −88.7 %

**5.1.3 Vertical cross-sections**

[revised manuscript text omitted]

---

## Author Response (AR2)

This paper describes a range of different set-ups for a high resolution atmospheric model simulating polynyas in the Laptev Sea. The set-up involves the use of "tiles" which are sub-grid scale parametrization of thin sea ice. The quality of the model and citations are generally fine. Polynyas are important regions with substantial ice production and very high heat fluxes, and are thus an important part of the Cryosphere.

The authors have made a decent attempt at re-writing the paper, and much of the overly detailed text has been removed, and a more general introduction has been added, compared to the first version.

For the main conclusion some new conclusions about the atmospheric boundary layer has been added, where the model actually should resolve processes in a better way than earlier model attempts. That the overall heat loss increases along with the total ice production when areas of thinner ice is added as a new lower boundary condition is indeed what is expected and does not contribute to an increased scientific understanding. I still find that the changes needed falls into the "major" category – as outlined below.

Major comments:

No discussion and explanation of increasing winds with thinner ice cover.
We have added an explanation in the text (section 5.1.2) and further added a figure (now Fig. 6d) that shows the increase in wind speed over the Anabar-Lena polynya during the case study and the surface pressure anomaly causing the increase.

Figures 4, 5, 6, and 10 have too small labels, legends and arrows.
Former Figures 4,5 and 10 have been enlarged and for all figures above we enlarged also the font size of the contour labels and legend. We further enlarged the arrows in the spatial plots.

MODIS based thin ice categories (Fig. 12) "explains" the thin ice tiles, and needs to be presented before the different simulations.
We agree and reordered the figures so that former Fig.12 is now Fig.3 and is referred to in section 2 where the model configuration is described.

"Realistic" assumptions versus ""reference simulation" needs a better explanation and introduction. Why is the most realistic simulation not the reference? I think this is caused be previous simulations with the same model, but it is not well explained.
We added more details on the "reference" simulation, which was configured as in Bauer et al. (2013) (as far as possible) so that a comparison with their results was possible. Further, at the same time it is the reference without a tile-approach for our sensitivity runs.

"Dense bottom water" formation is only added to the abstract and conclusion, but no results are present illustrating the possible consequences of a larger or smaller ice production, and no citations are given for such estimates either.
The corresponding text passages have been removed in the abstract and conclusion as we do not show direct results to this issue. We restricted our conclusion only to the Arctic sea-ice budget.

Detailed comments:

Abstract:

You should not use abbreviations in here, please spell out all words properly. And define them first time in the main text.
We removed all abbreviations from the abstract (and the conclusion, except CCLM).

The last sentence on "global thermohaline circulation" should be removed, this might serve as a motivation for the study, but no results are given, and there is no discussion on this aspect either.
We removed this text passage from the abstract, and the whole manuscript.

Line 8: ice production was estimated.
Corrected.

Introduction:

Page 1, Line 22: No citation on ocean circulation, just for atmosphere. Either remove "ocean circulation" or find a proper citation at least.
We added the references of Aksenov et al. (2011) and Rudels (1999) in relation to the Arctic circumpolar boundary current.

Page 3, line 7-8: Again the global importance has no citation, and suggests wishful thinking of your own favourite process ….
Removed.

Page 4, line 8: should be "proxy for dense water formation".
Corrected.

Lines 9 – 18: This is very detailed text that describes your model setup, it clearly belongs in the methods section.
We have moved this paragraph to the methods (section 2).

Section 2:
Page 6, Line 5: spell "outside" correctly.
Corrected.

Line 8: contradiction of terms here "open water areas are rarely free of ice" ….. What you mean is that there is usually a mix of open water and grease/frazil and thin solid ice in a polynya. The images supplied in the answer are clearly taken on calm days with little wind pushing ice away from the "up-wind" side.
We reformulated this sentence as it was misleading.

Section 4:
Please use "evaluation" instead of "verification", you can never, ever, verify that a model works perfectly, it is always a model of the real thing – nature. Otherwise this section is quite convincing, and very good that you have AWS observations available.
This is correct, we exchanged verification throughout the manuscript.

Section 5:
Page 12, line 2: the SIC is between 0% and 7%.
Corrected.

Line 21-22: The most interesting results to me is that the winds do increase when you use a thinner ice cover. This also seems to explain much of the changes in heat flux. Why is this

so? You do not discuss this.

We added an explanation in the manuscript (section 5.1.2) and again mention the explanation now in the discussion section in relation to the increased turbulent heat fluxes.

Figure 7: Letters H, E, L and K needs to be explained in the caption.

We added the missing explanations in the figure caption.

Page 16, line 1: Again the increasing wind – lacking explanation.

We added the explanation here again.

Line 4: "the location of clouds vary"

Corrected.

Page 17, line 7: The "realistic configuration" is not the "ref" model run. This is a bit odd. You made your choice though, so we can live with that. But – an explanation is need. It appears to come with the MODIS results in Figure 12. This is the background you depend upon for the model set-up, so it should be presented first.

An explanation has been added in section 2 to what reference refers to. We also moved Fig. 12 to Fig.3 and refer to it in section 2 as well.

Line 17: The "Bowen ratio" needs a citation.

We added the reference Bowen (1926).

Section 6

Page 18, Line 8 – 18: The polynya area is prescribed for the different model runs, and is thus model forcing. At least this is what I understand from your description. This is therefore not a result, and is part of the methods. This text should thus be moved to methods.

This is correct, we moved this section to the methods, it is now section 3.3.

Page 19, line 8 and line 25: The "realistic assumptions" here needs to be explained as noted above. A different word for this run might be better.

We explain the term "realistic assumption" now with respect to MODIS data or microwave sensors in general. We hope it is clear now what we mean with realistic.

Line 31-32: This increase in ice production and heat loss with thinner ice is NOT a result. It is zero order physics. If your model did not re-produce this basic physics you should be very worried. It is fine that it does, but it cannot be packed in as a result. There are of-course feed-back processes regulating if the response is linear, or not. This is the interesting bit, and a result.

We agree and have reformulated this sentence so that we now do not express this relationship as a result.

Section 7:

You need to implement the wind increase explanation (Page 20, line 3), and the "realistic assumption" (Page 20, line 10).

We added the explanation.

Page 21, line 26: Is there no ice production outside the polynya? This might be a consequence of the "offline post production" – but in nature there is of course continued growth outside the polynya. Must be mentioned.

There is ice production outside the polynya in our model simulations, which was also aggregated for the total ice production in the polynya masks. We added a sentence to clarify this.

Conclusion:

I suggest that all abbreviations should be removed from the Conclusion, one should be able to read it on it's own.
We removed the abbreviations except CCLM, which is a common term this model is referred to in the modelling community.

Page 22, Line 22: Please change to: " ….based on simulations with a regional atmospheric model (CCLM)". This is needed because many people read Conclusion first.
Changed.

Line 29: Add something after "realistic assumption" as noted with "major" points. Perhaps: "for the most realistic assumptions based on remote sensing of ice thickness".
We extended sentence with the suggested addition.

Page 23, line 1 – 3: Here you should clearly mention that a coupled models would be benefitial. Using a slab ocean + sea ice + atmosphere would produce it's own sea ice thickness based on the heat fluxes, as happens in nature.
We added a sentence as suggested.

**Anonymous Referee #1**

The authors addressed the raised issues in an appropriate way and the revised manuscript has strongly improved. Find below a few minor points which should be considered.

1. "About 29.1 km3 of ice production were estimated …" (Abstract, line 8): Unclear, add total winter ice production.
We added 'total winter ice production' to the sentence.

2. IP not explained (Abstract, line 9). Might be useful to add a table showing all abbreviation.
We removed all abbreviations from the abstract for a better reading as suggested by Reviewer 1.

3. Abstract, last sentence regarding impact of Laptev sea ice production: Quite a big step towards global THC, would add or replace with impact on Arctic sea ice budget / strength of Arctic sea ice decline.
We agree and we restrict our conclusion only to the Arctic sea-ice budget now.

4. Discussion of key role of vertical mixing (Page 3, Lines 6-8). Add references.
We have added the references of Bauch et al. (2009) and Johnsen and Polyakov (2001).

5. Discussion of Figure 7: "Then the contribution of the latent heat flux (E) increases by up to +10%, while the net longwave radiation L is reduced by up to 10%." (Page 17, lines 14-15): I do not understand this sentence and the given numbers. Comparing the reference run (black) and the realistic run (green) in Figure 7, the latent heat flux differs by a factor of 2 and is mainly responsible for the differences of the total balance; while the net longwave radiation fluxes hardly differ. Please clarify.
You are correct, this sentence is irritating as it is. We replaced the sentence by mentioning that the latent heat flux doubles in the realistic sensitivity run, which corresponds to the figure.

6. Discussion of Table 4 (section 6.2, page 19). How do you interpret the shown correlation coefficients? Why are the numbers larger for B10 and B00 when for all your simulations?

As the correlation is not too different (within the confidence interval of the C05 runs) we think that it is caused by the interpolation of the model results of B10 and B00 onto our C05 grid (as B10 and B00 used a different grid). Further, B10 and B00 used a different model version and domain, which might impact the wind speed and turbulent heat fluxes in the area.

[revised manuscript text omitted]